# Structure and function of the N-terminal domain of the human mitochondrial calcium uniporter

Youngjin Lee[1,2,‡], Choon Kee Min[1,3,†,‡], Tae Gyun Kim[1,2], Hong Ki Song[1,3], Yunki Lim[1,3], Dongwook Kim[1,3], Kahee Shin[1,3], Moonkyung Kang[4], Jung Youn Kang[1,2], Hyung-Seop Youn[1,2], Jung-Gyu Lee[1,2], Jun Yop An[1,2], Kyoung Ryoung Park[1,2], Jia Jia Lim[1,2], Ji Hun Kim[1,2], Ji Hye Kim[1,2], Zee Yong Park[1], Yeon-Soo Kim[4], Jimin Wang[2,5], Do Han Kim[1,3,*] & Soo Hyun Eom[1,2,6,**]

## Abstract

The mitochondrial calcium uniporter (MCU) is responsible for mitochondrial calcium uptake and homeostasis. It is also a target for the regulation of cellular anti-/pro-apoptosis and necrosis by several oncogenes and tumour suppressors. Herein, we report the crystal structure of the MCU N-terminal domain (NTD) at a resolution of 1.50 Å in a novel fold and the S92A MCU mutant at 2.75 Å resolution; the residue S92 is a predicted CaMKII phosphorylation site. The assembly of the mitochondrial calcium uniporter complex (uniplex) and the interaction with the MCU regulators such as the mitochondrial calcium uptake-1 and mitochondrial calcium uptake-2 proteins (MICU1 and MICU2) are not affected by the deletion of MCU NTD. However, the expression of the S92A mutant or a NTD deletion mutant failed to restore mitochondrial $Ca^{2+}$ uptake in a stable MCU knockdown HeLa cell line and exerted dominant-negative effects in the wild-type MCU-expressing cell line. These results suggest that the NTD of MCU is essential for the modulation of MCU function, although it does not affect the uniplex formation.

**Keywords** crystal structure; MCU; MCU domain-like fold; mitochondrial calcium uptake; uniplex
**Subject Categories** Membrane & Intracellular Transport; Structural Biology

## Introduction

Energized mitochondria take up large quantities of $Ca^{2+}$ through the putative tetrameric mitochondrial calcium uniporter (MCU) with a highly selective channel driven by a large electrochemical potential across the inner mitochondrial membrane (IMM) [1,2]. MCU is associated with many regulatory proteins such as mitochondrial calcium uptake-1 and mitochondrial calcium uptake-2 (MICU1 and MICU2), an MCU paralog (MCUb), and essential MCU regulator (EMRE), forming the mitochondrial calcium uniporter complex (uniplex) [3–7]. MICU1 and MICU2 contain two conserved EF-hand $Ca^{2+}$-binding domains that regulate the activity of MCU [3,4]. EMRE is a single-pass membrane protein with a highly conserved aspartate-rich tail and is required for interaction between MCU and MICU1/MICU2 [8]. Evidence also exists for a possible interaction between MCU and MCUR1. Overexpression of MCUR1 in HeLa cells increases mitochondrial $Ca^{2+}$ uptake, while its knockdown suppresses it [9]. Moreover, MCUR1 has recently been reported as a regulator for cytochrome c oxidase assembly [10].

MCU consists of two conserved transmembrane helices (TMs) connected by a 9-aa linker with a four-residue "DIME" motif flanked by an N-terminal domain (NTD) and C-terminal domain located within the mitochondrial matrix [11–13]. Computational modelling as well as biochemical experiments suggests that MCU tetramerization forms a highly $Ca^{2+}$-selective eight TM channel inside the IMM [5,11,12]. The two negatively charged residues "D" and "E" in the "DIME" motif are essential for MCU function, presumably providing $Ca^{2+}$-binding site(s) [11,12].

Balanced mitochondrial $[Ca^{2+}]$ is critical for the regulation of mitochondrial functions such as fission–fusion and ATP production [14]. Uncontrolled mitochondrial $Ca^{2+}$ overload caused by oncogenes and tumour suppressors can lead to the opening of the

1  School of Life Sciences, Gwangju Institute of Science and Technology (GIST), Gwangju, Korea
2  Steitz Center for Structural Biology, Gwangju Institute of Science and Technology (GIST), Gwangju, Korea
3  Systems Biology Research Center, Gwangju Institute of Science and Technology (GIST), Gwangju, Korea
4  Graduate School of New Drug Discovery & Development, Chungnam National University, Daejon, Korea
5  Department of Molecular Biochemistry and Biophysics, Yale University, New Haven, CT, USA
6  Department of Chemistry, Gwangju Institute of Science and Technology (GIST), Gwangju, Korea
   *Corresponding author. Tel: +82 62 715 2485; Fax: +82 62 715 3411; E-mail: dhkim@gist.ac.kr
   **Corresponding author. Tel: +82 62 715 2493; Fax: +82 62 715 2521; E-mail: eom@gist.ac.kr
   ‡These authors contributed equally to this work
   †Present address: New Drug Development Center, Daegu-Gyeongbuk Medical Innovation Foundation, Daegu, Korea

 

mitochondrial permeability transition pore (mPTP) with disruption of mitochondrial membrane potential [15]. Excess $Ca^{2+}$ entry in mitochondria has been associated with apoptosis and necrosis in many pathological states [16].

Overexpression or silencing of MCU causes muscular diseases [17]. Furthermore, knockdown of MCU results in energetic and developmental defects in *Trypanosoma brucei* [18] and zebrafish [19]. MCU knockdown can also cause embryonic lethality in a pure C57/BL/6 inbred mouse strain [20], although mild phenotypic changes have been reported in MCU knockout mouse models [21,22].

More research is needed to know the structural basis for the diverse functions of MCU. In this study, we present the crystal structures of MCU NTD in a novel fold. Our biochemical and functional characterization indicated that MCU NTD is essential for MCU activity, and NTD deletion or S92A mutation impair the function of MCU.

## Results

### Overall structure of MCU NTD and NTD-E

To elucidate the structural basis for MCU functions, we designed a set of human MCU truncation experiments for crystallographic and biochemical studies. We determined the first structure, at a 1.80 Å resolution, of the highly conserved NTD of MCU, corresponding to residues 75–165, encoded by exons 3 and 4, fused with the bacteriophage T4 lysozyme at the N-terminal end of the MCU NTD (Figs 1 and EV1A, Appendix Fig S1, Table 1). The T4 lysozyme fusion was used to enhance solubility and to phase a new crystal structure using molecular replacement. We also determined the structure of an extended version of the MCU NTD (MCU NTD-E), corresponding to residues 75–185, without the T4 lysozyme fusion at a 1.50 Å resolution (Figs 1 and EV1B, Table 1). The structure of MCU NTD-E was determined by molecular replacement using the MCU NTD structure as a template.

The structure of MCU NTD consists of one α-helix and six β-strands (Fig 1C) that form a central core, two highly conserved loops (L2 and L4) (Fig 1D and E) and one leucine-rich short α-helix (α2) (Fig 1F). Hydrophobic residues in the α2-helix ($_{140}$GIDLLL$_{145}$) stabilize the hydrophobic interior of MCU NTD through interactions among V108, I127 and F149 (Fig EV1C). MCU NTD-E has an additional α-helix (α3) and a C-terminal tail (Fig 1C, F and G). The L2 and L4 loops are stabilized by hydrogen bonds and hydrophobic

interaction formed by highly conserved L90, S92, R93, E95, E118, D119, I122 and one water molecule (W1) (Fig 1D and E). S92 located in the L2 loop forms a hydrogen bond with D119 located in L4 loop, stabilizing the local structure in these loops (Fig 1D). S92 is predicted as a potential CaMKII phosphorylation site [23], and we could expect that S92 phosphorylation induces conformational changes by breaking the hydrogen bonds in these loops and modulates MCU function [23,24]. The C-terminal tail in MCU NTD-E forms an extended coil structure and contains K180, a known ubiquitination and biotinylation site (Fig 1G) [25,26]. We hypothesize that ubiquitination of K180, which is located close to the MCU NTD core, is involved not only in ubiquitin-dependent proteasomal degradation, but also in the regulation of MCU function by inducing structural changes in MCU.

Of interest, we observed an unidentified electron density, which is predicted as a linear lipid-like molecule with 13–16 carbon atoms and modelled with a tetraethylene glycol molecule (Appendix Fig S2). The lipid-like molecule interacts with the residues in the L1 loop, two helices (α2 and α3) and C-terminus tail (Appendix Fig S2B).

### Identification of MCU domain-like fold

We identified MCU NTD as a novel fold and named it "MCU domain-like fold" superfamily based on the analysis of the Structural Classification of Proteins 2 (SCOP2) [27] (Fig 2). In a search for structures similar to MCU NTD using the Dali program [28] and CATH database [29], the ubiquitin-like (Ub) β-grasp fold (β-GF) ranked the highest in the Dali search (*Z*-score > 5.3) (Appendix Table S1), whereas the immunoglobulin (Ig)-like fold was the highest rank in the CATH database (SSAP score > 70) (Appendix Table S2). However, MCU NTD fold is different from β-GF and Ig-like fold. The MCU NTD core domain contains six β-strands and one α-helix in the following order: β4-β5-β6-α1-β1-β2-β3, in which each of the three β-strands forms two β-sheets (A- and B-sides) (Fig 2A and B). In contrast, the Ub core domain contains four β-strands and one α-helix in the following order: β3-β4-α1-β1-β2 (Fig 2C and D). The connection between β1-β2-α1 is highly conserved in all β-GFs [30]. However, in MCU NTD, β3 is inserted between β2 and α1, and one additional β-strand is inserted after α1 (Fig 2B). Although the two domains could be superposed with an RMSD of 2.73 Å based on their α1 and α1′-helices (Fig 2G), the directionality of the β-strands is different. Furthermore, when the domains were superposed based on the four β-strands, the α1- and α1′-helices of MCU NTD and Ub, respectively, are located in the opposite side (Fig 2H). On the other hand, the β2-macroglobulin core domain, one of Ig-like fold, consists of a 7- to

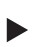

**Figure 1.   Overall structure of MCU NTD.**

A   Schematic diagram of the MCU sequence. MCU is composed of an N-terminal transit signal peptide (S), N-terminal domain (NTD), two transmembrane domains (TM1 and TM2), a "DIME" motif and two coiled-coils (CC). K180 is ubiquitination or biotinylation site [25,26].

B   Topology diagram of MCU in the IMM. Crystal structure of MCU NTD (orange) and MCU NTD-E (20-a.a. extension in magenta) is shown.

C   Overall structure of MCU NTD and MCU NTD-E. MCU NTD is composed of two helices (α1 and α2), six β-strands (β1–β6), and two conserved loops (L2 and L4). MCU NTD-E has an additional α-helix (α3) and C-terminal tail (magenta).

D   Top view of L2 and L4 loops (site A in C), showing the hydrogen bonding and hydrophobic interaction. Residues are shown in stick, one water molecule (W1) as red dots. Dashed lines (red) denote hydrogen bonds. The putative phosphorylation site, S92, is described in stick in the L2 loop.

E   Highly conserved L2 and L4 loops in MCU NTD by ConSurf analysis [61]. Residues represented in the L2 and L4 loops are coloured according to conservation analysis by ConSurf, using 250 MCU NTD homologues selected from the UniRef90 database.

F   C-terminal hydrophobic helical regions (α2 and α3; site B in C). α2- and α3-helices of the leucine-rich (LR) region are shown in orange and magenta, respectively.

G   Tail region (site C in C), showing the ubiquitination or biotinylation site K180 within MCU NTD-E.

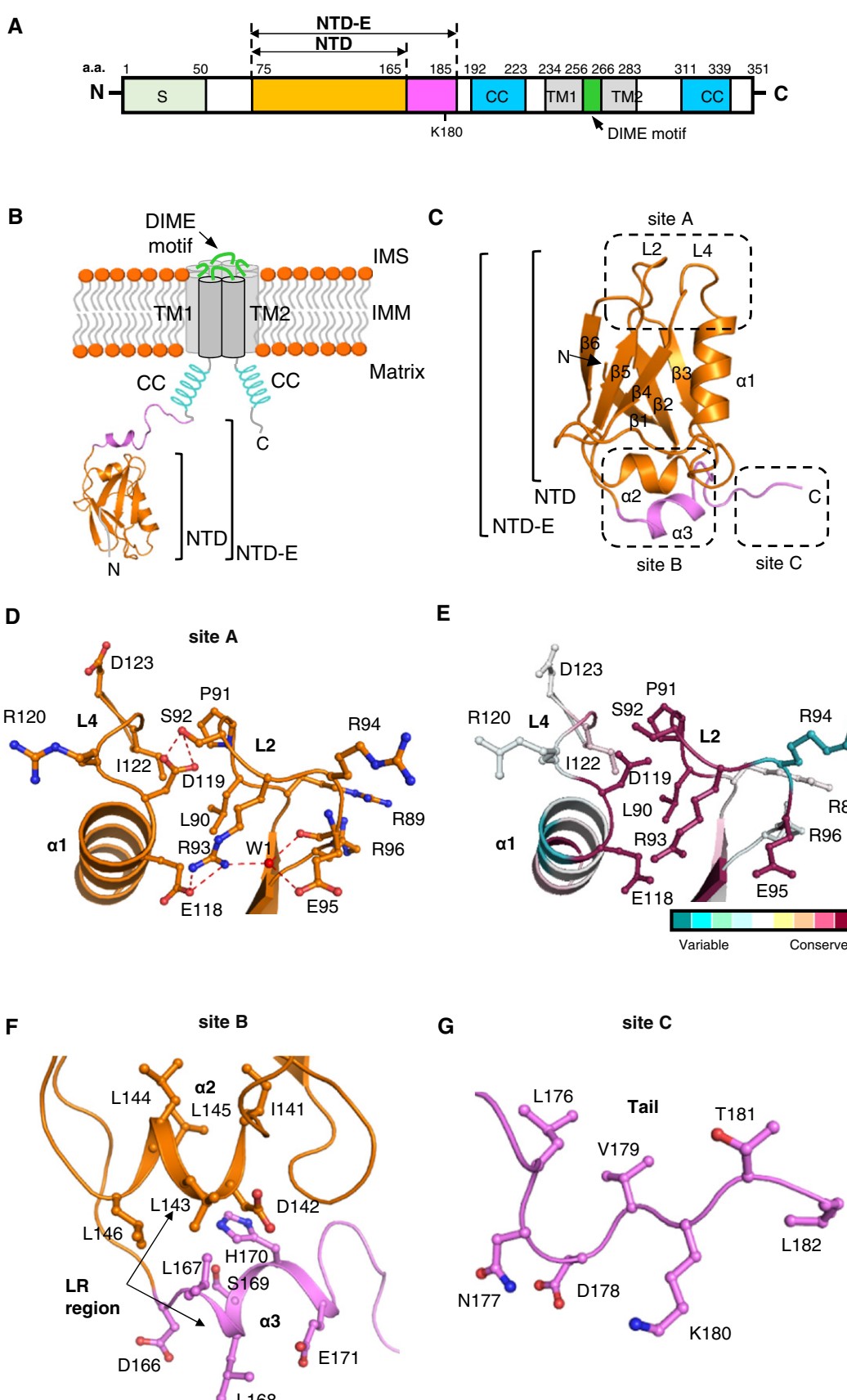

**Figure 1.**

**Table 1.  Data collection and refinement statistics.**

| | T4 lysozyme-MCU NTD<br>PDB ID: 4XSJ | T4 lysozyme-MCU NTD S92A<br>PDB ID: 5BZ6 | MCU NTD-E<br>PDB ID: 4XTB |
|---|---|---|---|
| **Data collection** | | | |
| Space group[a] | $P6_5$ | $P6_5$ | $P6_5$ |
| X-ray source[b] and detector | PAL-5C ADSC Q315r | PAL-5C ADSC Q315r | PAL-7A ADSC Q270 |
| Wavelength (Å) | 0.9795 | 0.9795 | 0.9793 |
| Unit cell: $a, b, c$ (Å) | 98.1, 98.1, 62.4 | 97.8, 97.8, 61.5 | 55.5, 55.5, 68.9 |
| $\alpha, \beta, \gamma$ (°) | 90.0, 90.0, 120.0 | 90.0, 90.0, 120.0 | 90.0, 90.0, 120.0 |
| Resolution range (Å)[c] | 50–1.80 (1.83–1.80) | 50–2.75 (2.80–2.75) | 50–1.50 (1.53–1.50) |
| $R_{merge}$[d] | 6.8 (54.6) | 12.3 (56.3) | 4.7 (51.5) |
| $I/\sigma I$ | 18.2 (3.2) | 6.8 (3.3) | 11.9 (3.6) |
| Completeness (%) | 99.5 (98.3) | 99.5 (100.0) | 99.4 (100.0) |
| Redundancy | 8.0 (6.3) | 4.7 (5.3) | 5.7 (5.6) |
| **Refinement** | | | |
| Resolution range (Å)[c] | 48.5–1.80 | 34.9–2.75 | 48.0–1.50 |
| No. reflections | 29581 | 8147 | 17594 |
| $R_{work}$[e] (%)/$R_{free}$ (%) | 12.7/19.0 | 16.4/23.5 | 14.0/17.6 |
| **No. atoms** | | | |
| Protein | 2008 | 2007 | 863 |
| Ligand | — | — | 13[f] |
| Ion ($SO_4^{2-}$) | 15 | 30 | — |
| Water | 431 | 41 | 143 |
| **B-factors (Å$^2$)** | | | |
| Protein | 27.0 | 33.7 | 18.2 |
| Ligand | — | — | 30.3 |
| Ion ($SO_4^{2-}$) | 40.6 | 56.7 | — |
| Water | 47.4 | 29.9 | 39.5 |
| **Model statistics** | | | |
| rmsd bond length (Å) | 0.010 | 0.014 | 0.010 |
| rmsd bond angles (°) | 1.33 | 1.67 | 1.64 |
| Ramachandran plot (%)<br>favoured/allowed/disallowed | 98.8/1.2/0 | 97.2/2.8/0 | 99.1/0.9/0 |

[a]$P6_5$-related MCU NTD/MCU NTD interactions are conserved in these three $P6_5$ crystal forms.
[b]Beamline 5C and 7A at Pohang Accelerator Laboratory (PAL) in South Korea.
[c]Values in parentheses are for highest-resolution shell.
[d]$R_{merge} = \sum_h \sum_i | I(h)_i - \langle I(h) \rangle | / \sum_h \sum_i I(h)_i$, where $I(h)$ is the intensity of reflection of $h$, $\sum_h$ is the sum over all reflections and $\sum_i$ is the sum over $i$ measurements of reflection $h$.
[e]$R_{work} = \sum_{hkl} ||F_o| - |F_c|| / \sum_{hkl} |F_o|$; 5% of the reflections were excluded for the $R_{free}$ calculation.
[f]An unidentified electron density was observed and modeled with a tetraethylene glycol molecule.

9-strand sandwich structure, including a Greek-key motif [31] (Fig 2E and F). Although the MCU NTD and β2-macroglobulin could be superposed with an RMSD of 4.9 Å, the orientation of the two β-sheets in the A-side is rotated by 42 degrees (Fig 2I).

**Potential protein–protein interaction interfaces of MCU NTD and NTD-E**

In the structure of MCU NTD-E, hydrophobic residues in the α2-helix (L143, L144 and L146) together with L167 and L168 in the α3-helix ($_{166}$DLLSHENA$_{173}$) form a hydrophobic surface surrounded by hydrophilic residues (namely D142, D166 and E171) (Fig 1F). This surface and C-terminus tail ($_{174}$ATLNNVKTL$_{182}$) were predicted as potential protein–protein interaction (PPI) interfaces by the Inter-ProSurf analysis [32] (Fig EV2A and C) and the consensus Protein–Protein Interaction Site Predictor (cons-PPISP) server [33] (Fig EV2B and C). In the crystals of T4 lysozyme-MCU NTD and MCU NTD-E, MCU NTDs form helical oligomers around the $6_5$ screw axis in a similar manner (Fig EV3A and B). In the crystal of MCU NTD-E, oligomers are stabilized by additional interactions through an extended C-terminal tail in a manner resembling domain swapping (Fig EV3B–D), and the interface in MCU NTD and the C-terminal tail

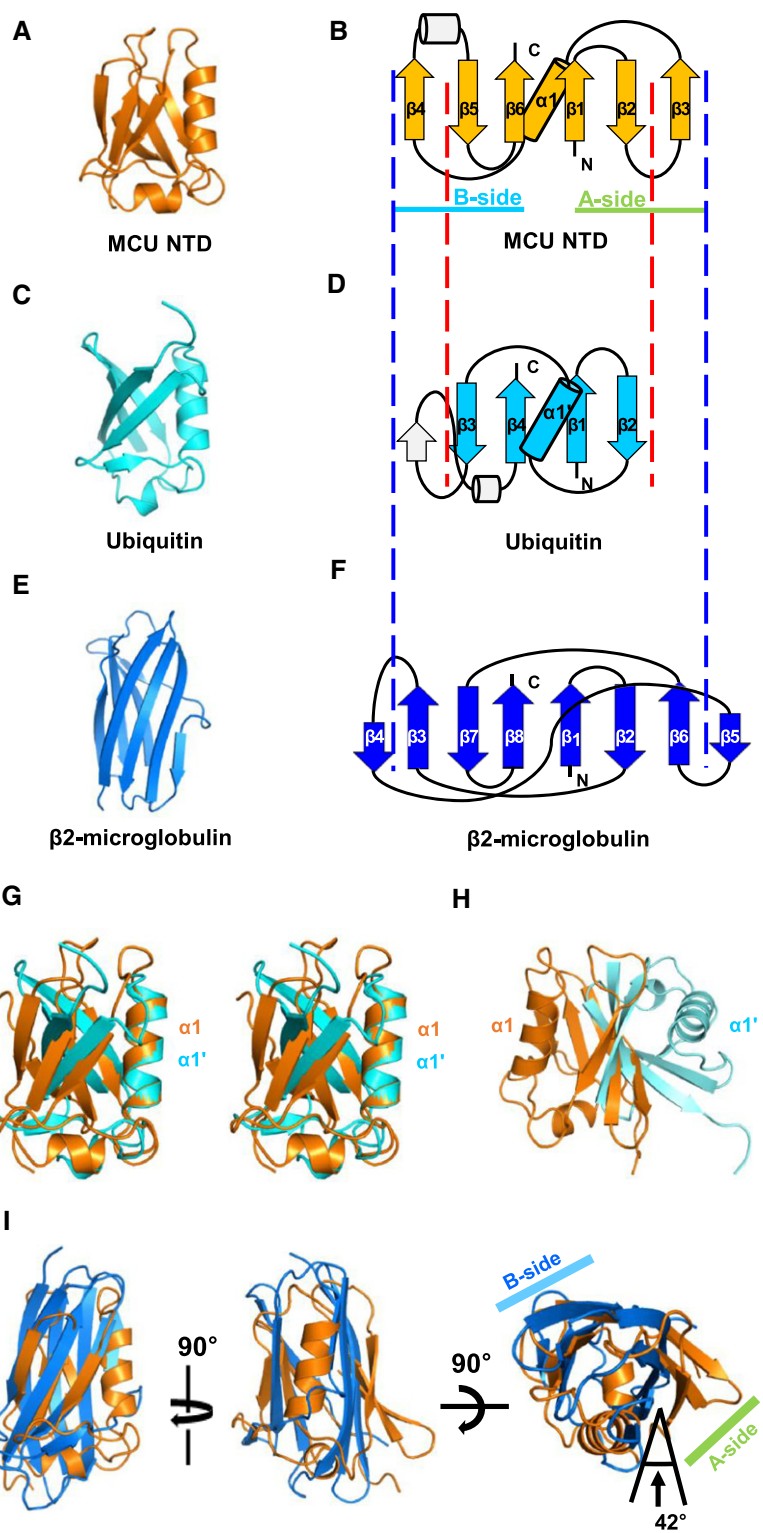

**Figure 2.   Topology comparison of MCU NTD, ubiquitin-like β-GF and immunoglobulin-like fold.**

A–F   Ribbon diagrams of MCU NTD (A), ubiquitin-like β-GF (Ub) (PDB code 1UBQ) (C) and Ig-like fold (β2-microglobulin were selected for comparison, chain F of PDB code 1IM3) (E). Topology diagrams of MCU NTD (B), Ub (D) and β2-macroglobulin (F). β-strands are represented by arrows, α-helices by cylinders and loops by lines. Each of the three β-strands of MCU NTD forms two β-sheets (A- and B-sides).
G   Stereo view of the superposition of MCU NTD (orange) and Ub (cyan) based on α1- and α1′-helices.
H   Superposition of MCU NTD (orange) and Ub (cyan) based on the four β-strands (red dashed lines) aligned between (B) and (D).
I   Superposition of MCU NTD (orange) and β2-microglobulin (blue) based on the central six β-strands (blue dashed lines) aligned between (B) and (F).

between subunits in the oligomer is consistent with the predicted PPI surface (Fig EV3E).

We also identified MCU NTD-Es oligomerized in solution by glutaraldehyde cross-linking assay in phosphate-buffered saline (PBS) and crystallization conditions (Appendix Fig S3). Thus, the results suggest that MCU NTD is involved in the oligomerization and interactions with the uniplex, although MCU NTD is not essential for the formation of the uniporter complex (see below).

**Overexpression of MCU$_{\Delta NTD}$ exerts a dominant-negative function without alteration of the uniplex assembly**

MCU forms homo-oligomers as well as hetero-oligomers (with MCUb) [5] and interacts with various regulatory proteins, including MICU1 and MICU2, in the intermembrane space forming a uniplex [8,34,35]. To investigate the functional role of the MCU NTD, we deleted this domain by generating an MCU mutant lacking residues 75–165 (MCU$_{\Delta NTD}$). Subcellular localization of MCU$_{\Delta NTD}$ within the mitochondria was confirmed by co-expressing C-terminally GFP-tagged MCU$_{\Delta NTD}$ and DsRed-Mito (Fig EV4C). Expression of MCU$_{\Delta NTD}$ did not affect the endogenous expression of MCU or that of the regulatory proteins (Fig EV4A, B and D).

To examine the structural effect of MCU NTD deletion, we performed circular dichroism (CD) analysis for MCU$_{\Delta NTD}$ (Appendix Fig S4A). The CD spectrum of MCU$_{\Delta NTD}$ showed that the canonical α-helical pattern and the α-helical content were well matched with the predicted values, which were approximately 66 and 61%, respectively (Appendix Fig S4A and B), suggesting that MCU$_{\Delta NTD}$ is properly folded.

To determine whether MCU NTD is involved in the assembly of MCU-containing uniplex, we performed a co-immunoprecipitation assay and blue native polyacrylamide gel electrophoresis (BN–PAGE). As shown in Fig 3A and B, both MCU$_{WT}$ and MCU$_{\Delta NTD}$ expressed in HeLa cells were co-precipitated with MCU$_{\Delta NTD}$, suggesting that MCU NTD is not essential for MCU oligomerization (Appendix Fig S5A and B). In addition, co-immunoprecipitation assays using Flag-tagged MCU$_{WT}$ and MCU$_{\Delta NTD}$ showed that

MCU$_{\Delta NTD}$ binds both MICU1 and MICU2 as MCU$_{WT}$ does (Fig EV4E and Appendix Fig S5C). BN–PAGE using digitonin-solubilized mitochondria isolated from wild-type and MCU knockdown (MCU-KD) HeLa cells overexpressing Flag-tagged MCU$_{\Delta NTD}$ showed that MCU$_{\Delta NTD}$ migrates at an apparent molecular weight of 440 kDa (Figs 3C and 4A and B, Appendix Fig S5D). Since it has been reported that uniplex migrates at an apparent molecular weight of approximately 480 kDa in BN–PAGE [8], the shift of MCU$_{\Delta NTD}$ complex correlates well with the size of the MCU NTD deletion, suggesting that the deletion did not alter the assembly of the MCU-containing uniplex.

We next investigated whether the deletion of MCU NTD from MCU affects mitochondrial Ca$^{2+}$ uptake in intact cells. To monitor both cytosolic and mitochondrial [Ca$^{2+}$] simultaneously, we used genetically encoded Ca$^{2+}$ indicators that we modified from the previously developed genetically encoded Ca$^{2+}$ indicators for optical imaging (GECOs) such that they specifically targeted mitochondria [36]. After confirming the expression of both the mitochondria-targeting green-GECO (mito-GGECO) and red-GECO (RGECO) in HeLa cell mitochondria and cytosol, we monitored the changes of fluorescence intensity induced by simultaneous stimulation of the two compartments (Appendix Fig S6). Consistent with an earlier report [12], MCU overexpression reduced the amplitude of the cytosolic [Ca$^{2+}$] peak evoked by 100 μM histamine (Fig 3F and G), while enhancing mitochondrial Ca$^{2+}$ uptake (Fig 3D and E). However, when we overexpressed MCU$_{\Delta NTD}$, not only was the increase of mitochondrial Ca$^{2+}$ uptake in MCU$_{WT}$-overexpressing HeLa cells abrogated, but mitochondrial Ca$^{2+}$ uptake was also reduced relative to the control experiment (Fig 3D and E). In addition, the cytosolic [Ca$^{2+}$] peak was enhanced in the MCU$_{\Delta NTD}$-overexpressing cells relative to MCU$_{WT}$, possibly due to reduced mitochondrial Ca$^{2+}$ buffering (Fig 3F and G). To examine the possibility that an overexpression of MCU$_{\Delta NTD}$ affects the driving force for Ca$^{2+}$ uptake, the mitochondrial membrane potential was investigated using tetramethylrhodamine methyl ester (TMRM). The results showed that TMRM loading was unaffected by MCU$_{\Delta NTD}$ overexpression (Fig 3H and I). These data suggested that MCU$_{\Delta NTD}$

**Figure 3.  MCU$_{\Delta NTD}$ overexpression has a dominant-negative effect on mitochondrial Ca$^{2+}$ uptake.**

A    Co-immunoprecipitation of MCU$_{WT}$-Flag or MCU$_{\Delta NTD}$-Flag with MCU$_{WT}$-GFP. HeLa cells were transiently co-transfected with MCU$_{WT}$-GFP and MCU$_{WT}$-Flag/MCU$_{\Delta NTD}$-Flag. MCU$_{\Delta NTD}$ migrated farther than MCU$_{WT}$, indicating an apparent molecular weight difference of 9 kDa in SDS–PAGE (see also Fig EV4A). MCU$_{WT}$-GFP with MCU$_{WT}$-Flag or MCU$_{\Delta NTD}$-Flag was precipitated from the cell lysates with an anti-GFP antibody. The precipitates were separated on SDS–PAGE and immunoblotted with the antibodies indicated.

B    Co-immunoprecipitation of MCU$_{\Delta NTD}$-Flag with MCU$_{\Delta NTD}$-GFP. HeLa cells were transiently co-transfected with MCU$_{\Delta NTD}$-Flag and MCU$_{\Delta NTD}$-GFP. MCU$_{\Delta NTD}$-GFP with MCU$_{\Delta NTD}$-Flag was precipitated from cell lysates with an anti-GFP antibody. The precipitates were separated on SDS–PAGE and immunoblotted with the indicated antibodies.

C    HeLa cells were transiently transfected with MCU$_{WT}$-Flag or MCU$_{\Delta NTD}$-Flag. After isolation and solubilization of crude mitochondria, the lysates were subjected to BN–PAGE and immunoblotted with anti-Flag and anti-MCU antibodies to detect ectopic MCU$_{WT}$-Flag, MCU$_{\Delta NTD}$-Flag and endogenous MCU. MCU$_{WT}$ and MCU$_{\Delta NTD}$ were detected at apparent molecular weights of 480 and 440 kDa, respectively. The shift shown in MCU$_{\Delta NTD}$ complex correlated with the difference in molecular weight, if it is assumed to be a tetramer. Flavoprotein subunit of complex I is used as a loading control for each mitochondrial fraction.

D–G  Simultaneous measurements of mitochondrial (D, E) and cytosolic (F, G) Ca$^{2+}$ transients evoked by 100 μM histamine in HeLa cells overexpressing MCU$_{WT}$ or MCU$_{\Delta NTD}$. F$_0$ is initial fluorescence intensity. F and F$_{max}$ indicate fluorescence intensity at each time point and maximal fluorescence intensity after the stimulation, respectively. (F$_{max}$–F$_0$)/F$_0$ indicates the maximal Ca$^{2+}$ concentration evoked by the stimulation (mean ± SEM, n = 12, *$P < 0.05$ versus empty vector-transfected cells. #$P < 0.05$ versus MCU$_{WT}$ vector-transfected cells). Unpaired two-sided Student's *t*-test was used to calculate statistical significance.

H, I  The mitochondrial membrane potential was monitored based on TMRM fluorescence intensity. After loading of TMRM, FCCP (carbonyl cyanide p-trifluoro-methoxyphenylhydrazone), an uncoupler of oxidation phosphorylation, was rapidly applied to disrupt the mitochondrial membrane potential established by the respiratory chain (ΔΨ). Fluorescence intensities of each group were recorded at each time point using a LSM 700 confocal laser-scanning microscope. The value of TMRM loading (relative intensity) was calculated by dividing each fluorescence intensity of TMRM measured with MCU$_{WT}$ or MCU$_{\Delta NTD}$ vector by that of TMRM measured with empty vector (mean ± SEM, n = 7).

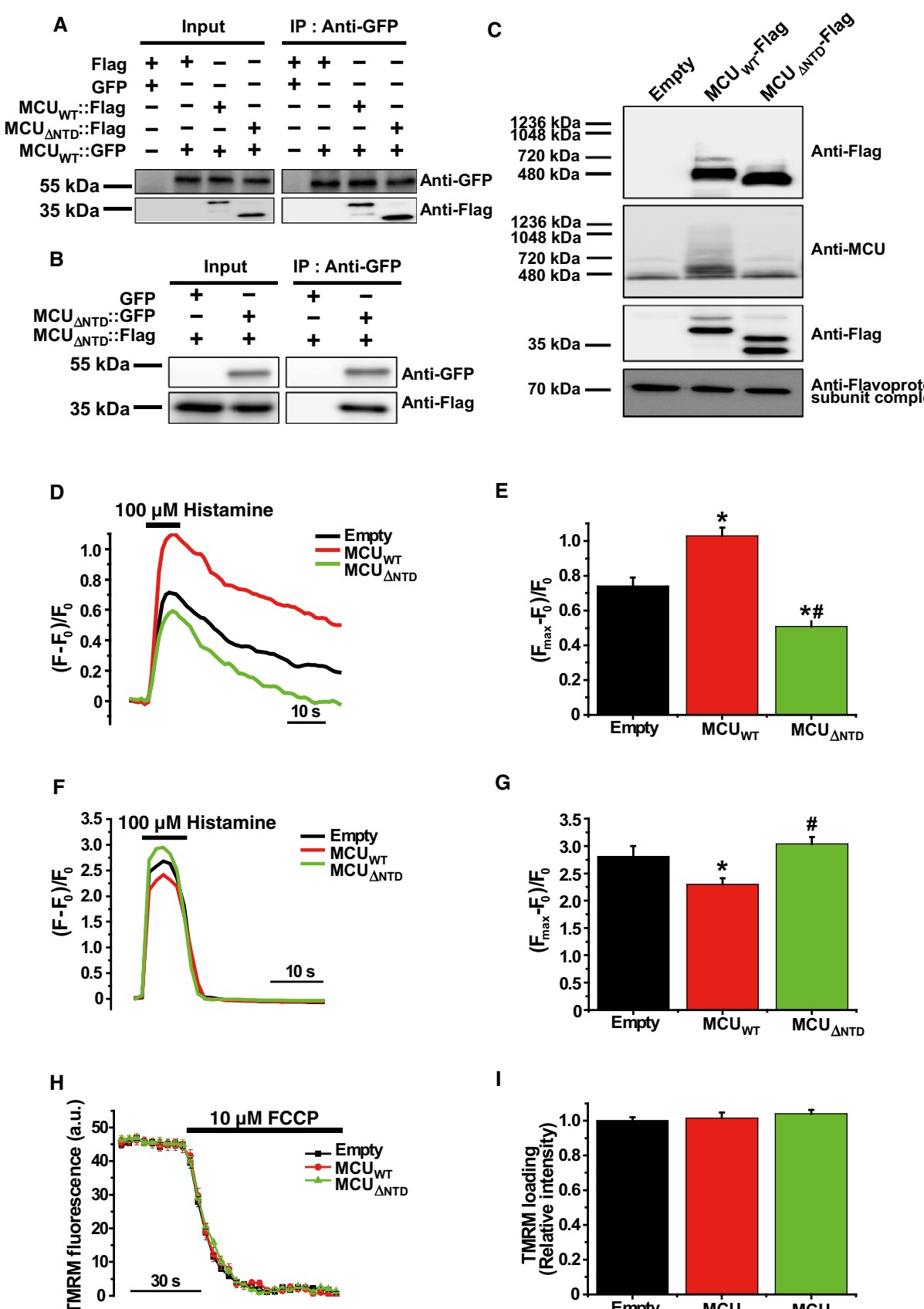

**Figure 3.**

exerts a dominant-negative effect on mitochondrial $Ca^{2+}$ uptake after incorporation into some uniplexes.

### Rescue of MCU$_{\Delta NTD}$ in stable MCU-KD cells could not fully restore mitochondrial $Ca^{2+}$ uptake

To determine whether MCU$_{\Delta NTD}$ is intrinsically $Ca^{2+}$ impermeable or presents a reduced channel activity, we performed a rescue experiment after generation of stable MCU-KD HeLa cells through transduction of a lentiviral vector carrying short hairpin RNA (shRNA) specifically targeting MCU. Consistent with the previous results (Fig 3F and G), cytosolic $[Ca^{2+}]$ peak was increased in MCU-KD HeLa cells (Fig 4E and F). When we monitored the changes in mitochondrial $[Ca^{2+}]$ in HeLa cells, MCU-KD abrogated histamine-evoked mitochondrial $Ca^{2+}$ uptake, but the expression of control shRNA (shCon) did not (Fig 4A, C and D). Expression of MCU$_{WT}$ with nine silent point mutations in the shRNA target sequence that did not affect the amino acid sequence restored stimulation-induced mitochondrial $Ca^{2+}$ uptake in stable MCU-KD HeLa cells (Fig 4A, C and D). Although MCU$_{\Delta NTD}$ expression also led to mitochondrial $Ca^{2+}$ uptake, the amplitude of the response was only about half of that observed with MCU$_{WT}$ without alteration of driving force ($\Psi$) (Fig 4G and H), suggesting that MCU$_{\Delta NTD}$ may still function as a $Ca^{2+}$ channel, but $Ca^{2+}$ influx was lower than that of MCU$_{WT}$.

### MCU S92A mutation reduces mitochondrial $Ca^{2+}$ uptake

Several post-translational modifications in MCU NTD have been reported including a putative phosphorylation site for CaMKII on S92 and a biotinylation site on K180 [23,26]. In order to investigate whether the post-translational modifications at S92 and K180 are crucial for MCU-mediated mitochondrial $Ca^{2+}$ uptake, we generated MCU$_{S92A}$ and MCU$_{K180A}$ mutants. Then, we monitored mitochondrial $Ca^{2+}$ uptake in MCU$_{S92A}$- or MCU$_{K180A}$-rescued MCU-KD HeLa cells. The results showed that mitochondrial $Ca^{2+}$ uptake was only impaired in MCU$_{S92A}$-rescued cells, but not changed in MCU$_{K180A}$-rescued cells (Fig 5C and D).

To examine the structural effect of these mutations, we performed CD analysis using MCU NTD-E S92A and K180A constructs (Appendix Fig S4C). We identified no changes in secondary structure contents, suggesting that S92A and K180A mutants are properly folded. In addition, BN–PAGE showed that both mutants had the same apparent molecular weight of 480 kDa

as WT had (Fig 5A). Furthermore, co-immunoprecipitation assays showed that MCU mutants are co-precipitated with MCU$_{WT}$ (Fig 5B) and there were no defects in binding with both MICU1 and MICU2 as MCU$_{WT}$ does (Fig EV4E). These results suggest that MCU$_{S92A}$ and MCU$_{K180A}$ do not alter MCU folding, oligomerization and assembly of MCU-containing uniplex. Taken together, S92 and K180 residues are not involved in uniplex formation, but among them only the S92 residue is crucial for mitochondrial $Ca^{2+}$ uptake activity.

### Crystal structure of NTD$_{S92A}$ suggests a conformational change in the mutant

To elucidate the structural basis for the impaired activity of MCU$_{S92A}$ as shown above, we determined the 3D structure of MCU NTD$_{S92A}$ with N-terminus T4 lysozyme fusion at 2.75 Å resolution (Fig 6 and Table 1). Overall structures of the NTD$_{WT}$ and NTD$_{S92A}$ are similar with a RMSD of 0.60 Å (Fig 6A). Interestingly, we found that the NTD$_{S92A}$ induced a conformational change in the L2-L4 loops (Fig 6). In the structure of NTD$_{WT}$, hydrogen bonding between S92 and D119 maintains *trans*-conformation of the P91, which is located next to S92 (Fig 6B). On the contrary, NTD$_{S92A}$ breaks the S92-D119 hydrogen bonding and induces conformational change to *cis*-form of P91 (Fig 6C). Overall, L2 loop conformation in the NTD$_{S92A}$ moves away from L2 loop of NTD$_{WT}$ at a distance of $C_\alpha$ atom of 5.6 Å (Fig 6A and D). Sequentially, side chain of R93 forming hydrogen bond with E118 in the NTD$_{WT}$ moves up to the position of S92 and makes a new hydrogen bonding with D119 in the NTD$_{S92A}$ (Fig 6D). In addition, L90 interacting with hydrophobic residues, V88, L115, I122, V125 and I153, in NTD$_{S92A}$ moves away from L90 of NTD$_{WT}$ at a distance of 2.5 Å (Fig 6D). Thus, we suggest that a conformational change in L2-L4 loops of the NTD$_{S92A}$ impairs the mitochondrial $Ca^{2+}$ uptake activity.

## Discussion

The uniplex reportedly consists of MCU and regulatory proteins, including MICU1, MICU2 and EMRE [6–8]. The present results show that MCU NTD is not crucial for the uniplex assembly. Nevertheless, MCU NTD might be important for the interaction with other regulators or intra-molecular interaction for mitochondrial $Ca^{2+}$ uptake, since MCU NTD has putative PPI surface, and the $Ca^{2+}$ uptake activity of MCU$_{\Delta NTD}$ and MCU$_{S92A}$ was significantly reduced in the

---

**Figure 4.  MCU$_{\Delta NTD}$ expression in stable MCU-KD HeLa cells does not fully restore mitochondrial $Ca^{2+}$ uptake.**

A    Expression of MCU$_{WT}$-Flag and MCU$_{\Delta NTD}$-Flag in stable MCU-KD HeLa cells. Twenty micrograms of each sample were subjected to SDS–PAGE and detected with the indicated antibodies. Anti-MCU-C antibody was raised against a synthetic peptide (residues, 328–351) of MCU and detected both MCU$_{WT}$ and MCU$_{\Delta NTD}$, while commercially available MCU antibody (Sigma) was not able to detect MCU$_{\Delta NTD}$, because its epitope is the residues 47–152.

B    The solubilized mitochondrial fraction from each sample was subjected to BN–PAGE and immunoblotted with anti-Flag antibody.

C–F  Mitochondrial (C, D) and cytosolic (E, F) $Ca^{2+}$ transients evoked by 100 µM histamine were quantified by measuring the peak amplitudes of the traces, simultaneously. $F_0$ is initial fluorescence intensity. F and $F_{max}$ indicate fluorescence intensity at each time point and maximal fluorescence intensity after stimulation, respectively. $(F_{max}–F_0)/F_0$ indicates the maximal $Ca^{2+}$ concentration evoked by stimulation (mean ± SEM, n = 12–18, *$P < 0.05$ versus control shRNA-expressing cells. #$P < 0.05$ versus stable MCU-KD cells. §$P < 0.05$ versus MCU$_{WT}$-rescued stable MCU-KD cells). Unpaired two-sided Student's *t*-test was used to calculate statistical significance.

G, H  The mitochondrial membrane potential was monitored based on TMRM fluorescence intensity. After loading of TMRM, FCCP was rapidly applied to disrupt the mitochondrial membrane potential established by $\Delta\Psi$. Fluorescence intensities of each group were recorded at each time point using a LSM 700 confocal laser-scanning microscope. The value of TMRM loading (relative intensity) was calculated by dividing each fluorescence intensity of TMRM measured with MCU$_{WT}$ or MCU$_{\Delta NTD}$ vector by that of TMRM measured with empty vector (mean ± SEM, n = 8).

---

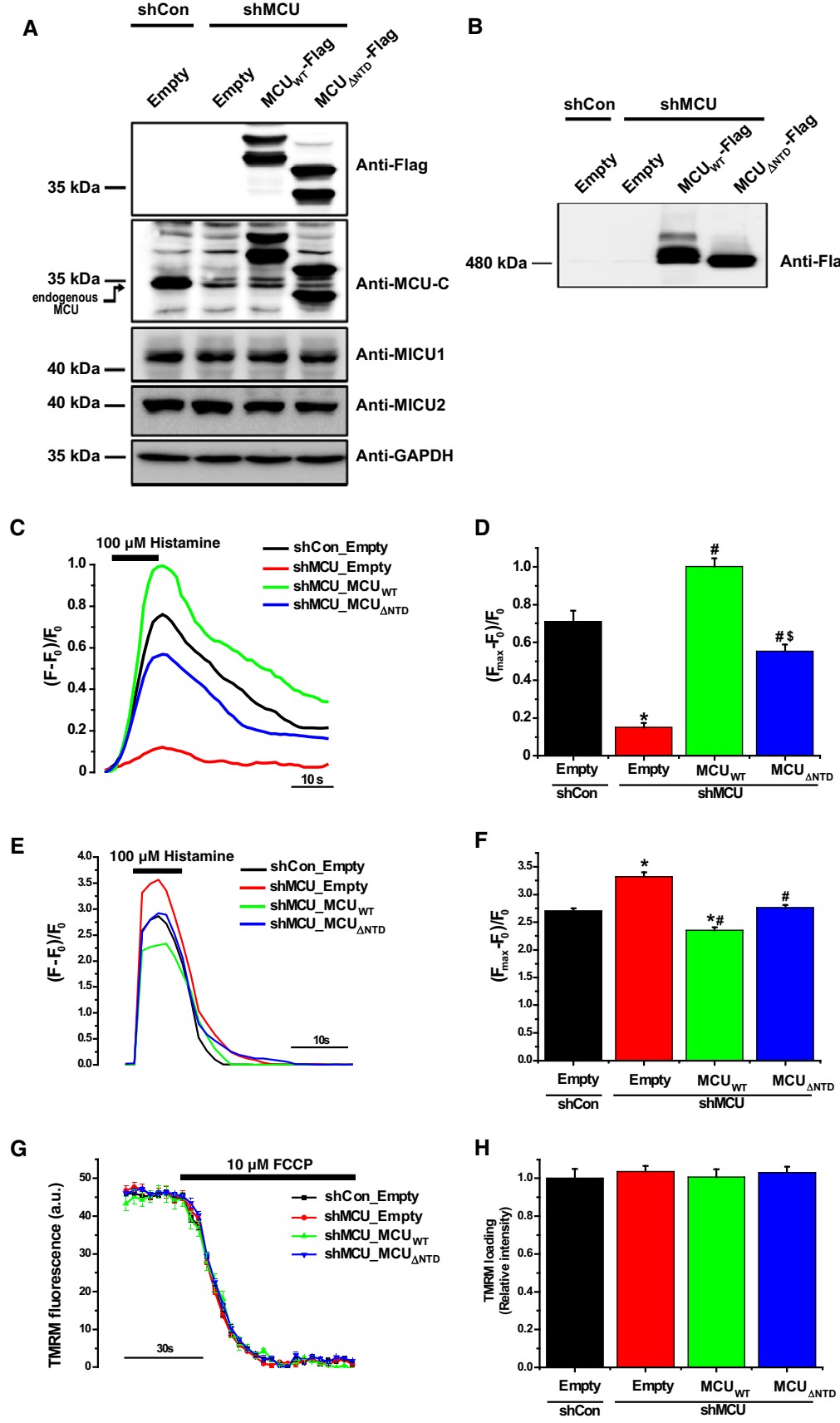

**Figure 4.**

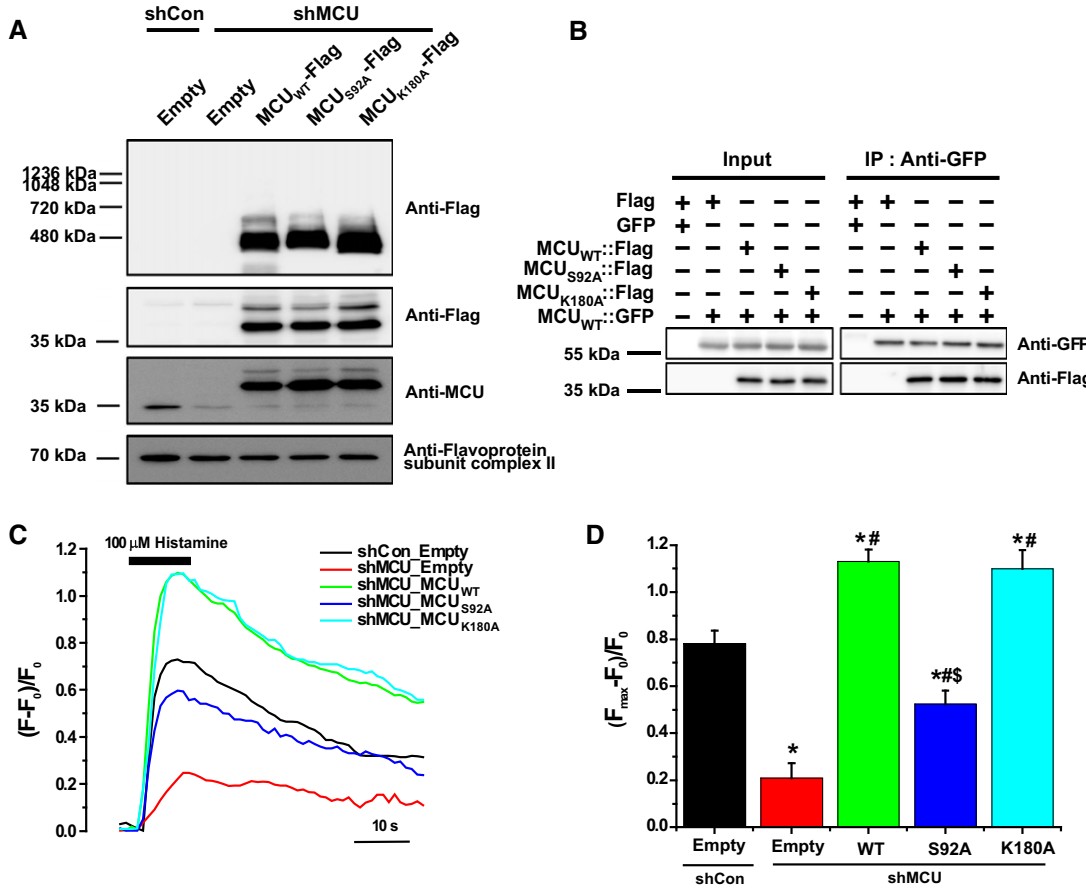

**Figure 5.  Substitution of Ser92 to Ala in MCU NTD reduced mitochondrial Ca$^{2+}$ uptake activity.**

A    Expression of MCU$_{WT}$-Flag, MCU$_{S92A}$-Flag and MCU$_{K180A}$-Flag in stable MCU-KD HeLa cells. The solubilized mitochondrial fraction from each sample was subjected to BN–PAGE and immunoblotted with anti-Flag antibody. The lysates of each sample were subjected to SDS–PAGE and detected with the indicated antibodies. Flavoprotein subunit of complex I is used as a loading control of each mitochondrial fraction.

B    Interaction of MCU mutants with MCU$_{WT}$ was not altered. After co-expression of MCU$_{WT}$ and MCU mutants, co-immunoprecipitation was performed. The precipitates were separated by SDS–PAGE and immunoblotted with the indicated antibodies.

C, D  Mitochondrial Ca$^{2+}$ uptakes evoked by 100 μM histamine were quantified by measuring the peak amplitudes of the traces. $F_0$ is initial fluorescence intensity. F and $F_{max}$ indicate fluorescence intensity at each time point and maximal fluorescence intensity after the stimulation, respectively. $(F_{max}-F_0)/F_0$ indicates the maximal Ca$^{2+}$ concentration evoked by the stimulation (mean ± SEM, $n$ = 13–15, *$P$ < 0.05 versus control shRNA-expressing cells. $^{\#}P$ < 0.05 versus stable MCU-KD cells. $^{§}P$ < 0.05 versus MCU$_{WT}$-rescued stable MCU-KD cells). Unpaired two-sided Student's *t*-test was used to calculate statistical significance.

present study. Although the fact that MCUR1 is a regulator for mitochondrial Ca$^{2+}$ uptake by interaction with MCU is controversial [9,10], we identified that MCU NTD directly interacts with MCUR1 (Fig EV5). However, how the interaction of MCU NTD with MCUR1 affects MCU function remains to be elucidated.

It has been reported that high [Ca$^{2+}$] is required for the Ca$^{2+}$ transmission between endo/sarcoplasmic reticulum (E(S)R) and mitochondria, since MCU has very low affinity for Ca$^{2+}$ [37,38]. Several tethering proteins support the close apposition between the subcellular organelles [39,40] and the interaction between Ca$^{2+}$ releasing channels (IP3R and RyR) in E(S)R and Ca$^{2+}$ pathways (VDACs) across the mitochondrial outer membrane could mediate high [Ca$^{2+}$] microdomains around the uniplex [41–43]. Considering each RyRs and VDACs clustered in the microdomain [43–45], it opens the possibility of MCU clustering at the microdomain for efficient permeation across the mitochondrial inner membrane. In fact, we observed that MCU NTD forms an oligomer in the

crystals and in the solution (Fig EV3A and B, Appendix Fig S3). Furthermore, MCU NTDs form helical oligomers in a similar manner to both the crystals of T4 lysozyme-MCU NTD and MCU NTD-E, suggesting that a crystal packing interface between MCU NTDs is not random (Fig EV3A and B). In addition, the interface in MCU NTD oligomer is consistent with the predicted PPI surface (Fig EV3E), suggesting that the interface observed in the crystal packing and PPI prediction in MCU NTD might be involved in the clustering of uniplexes at the microdomain. Thus, the deletion of MCU NTD could cause deprivation of spatial organization of the uniplex at the microdomain resulting in diminished mitochondrial Ca$^{2+}$ uptake (Figs 3 and 4). However, the detailed molecular mechanisms for the uniplex clustering at the microdomain will await for the future studies such as resolving the 3D structures of the oligomers involved.

Post-translational modifications could alter ion channel activity directly or indirectly by attachment of biochemical functional

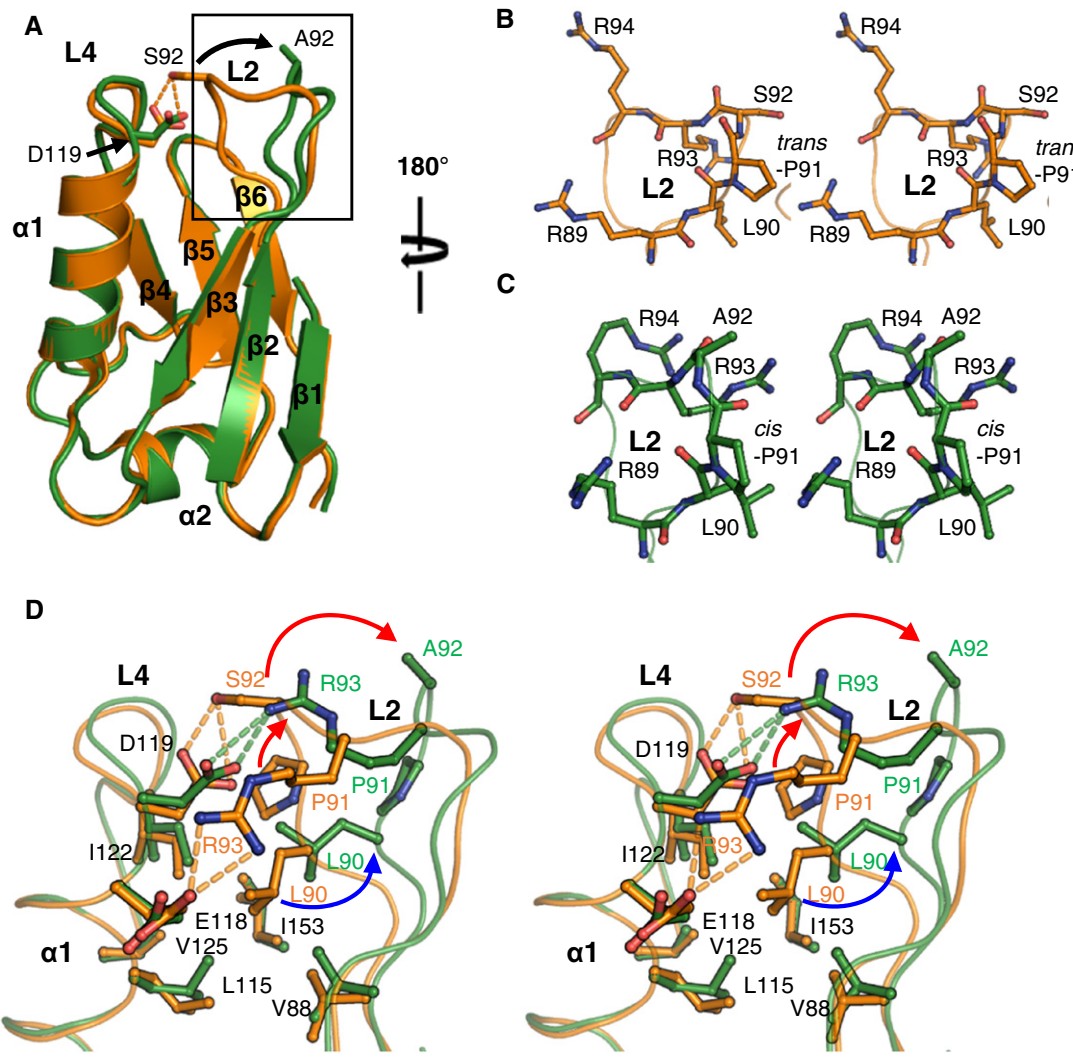

**Figure 6.   Structural comparison of MCU NTD$_{WT}$ and NTD$_{S92A}$.**

A     Superposition of the overall structure of MCU NTD$_{WT}$ (orange) and NTD$_{S92A}$ (green). The arrow (black) in the box indicates the difference of L2 loop between NTD$_{WT}$ and NTD$_{S92A}$. Dashed lines (orange) denote hydrogen bonds.

B, C  Stereoview of L2 loop of MCU NTD$_{WT}$ (B) and NTD$_{S92A}$ (C). P91 in the L2 loop describes two forms, *trans* (B) or *cis* (C).

D     Stereoview of conformational changes of the MCU NTD$_{WT}$ and NTD$_{S92A}$. Dashed lines denote hydrogen bonds in NTD$_{WT}$ (orange) and NTD$_{S92A}$ (green). The arrows indicate the movement of residues forming hydrogen bonding (red) or hydrophobic interaction (blue), respectively.

groups [23,46–48]. S92 in the L2 loop was predicted as a putative phosphorylation site for CaMKII, and its phosphorylation could modulate MCU function [23]. Our study showed that the highly conserved S92 in L2 loop and D119 in L4 loop form a hydrogen bond and stabilize the loop structures (Fig 1D and E). In addition, the NTD$_{S92A}$ induced a conformational change in L2–L4 loops and impaired mitochondrial Ca$^{2+}$ uptake activity (Figs 5C and D, and 6), suggesting the possibility that S92 phosphorylation induces structural change and charge distribution in L2–L4 loops resulting in the modulation of the MCU function.

In summary, we determined the crystal structure of MCU NTD which forms a novel fold. MCU NTD appears to be essential for the regulation of mitochondrial Ca$^{2+}$ uptake through interaction with regulator(s) and/or modulation of post-translational modifications. Our results provide a framework for future studies, investigating

how MCU controls Ca$^{2+}$ permeation across the IMM via the uniplex and the possible roles of the post-translational modifications.

# Materials and Methods

### Cloning MCU constructs

The sequences encoding MCU (NP_612366) residues 75–165 (MCU NTD) and MCU residues 75–185 (MCU NTD-E) were amplified using polymerase chain reaction (PCR) from human oral squamous carcinoma YD-10B cDNA. For *in vitro* binding assays, each of MCU NTD sequence was first cloned into a modified pET28a vector (Novagen) containing N-terminal 6× His (His$_6$)-tobacco etch virus (TEV) or His$_6$-maltose binding protein (MBP)-TEV. Each sequence was also

cloned into modified pET41a vector (Novagen) containing glutathione S-transferase (GST), which altered the thrombin site for a TEV protease site. For crystallographic studies, MCU NTD$_{WT}$ and the mutant, NTD$_{S92A}$, were further cloned into modified pET21a vector (Novagen), which includes N-terminal His$_6$-bacteriophage T4 lysozyme (residues 2–161). The T4 lysozyme was designed with triple mutations to prevent both cysteine residue oxidation (C54T/C97A) [49] and bacterial cell lysis (D20N) upon protein expression [50]. MCU NTD-E was finally cloned into pColdII vector (TaKaRa), which contains N-terminal His$_6$.

For the expression of MCU and MCU$_{\Delta NTD}$ in HeLa and HEK-293 FT cells, MCU$_{WT}$, MCU$_{\Delta NTD}$, MCU$_{S92A}$ and MCU$_{K180A}$ were cloned into the pHM6 (Roche), pCS4 (Roche) and pEGFP-N2 (Clontech) vectors. The pHM6 vector system contains an N-terminal HA and C-terminal His$_6$. The pCS4 and pEGFP-N2 vectors encode C-terminal 3× Flag and green fluorescent protein (GFP), respectively.

### Expression and purification of MCU constructs

T4 lysozyme-MCU NTD$_{WT}$ and NTD$_{S92A}$ were expressed in *Escherichia coli* strain BL21-CodonPlus (DE3). Transformed cells were cultured in Luria–Bertani medium containing 100 µg ml$^{-1}$ ampicillin at 37°C. After the addition of 0.5 mM isopropyl-β-D-thiogalactopyranoside (IPTG) (Goldbio), the cells were incubated at 20°C for an additional 20 h. They were then harvested by centrifugation at 4,000 × g for 20 min, resuspended in lysis buffer containing 20 mM Tris–HCl (pH 8.0), 500 mM NaCl, 10 mM imidazole, 1 mM PMSF and 1 mM β-mercaptoethanol, lysed by sonication and again centrifuged for 1 h at 14,000 × g. The resultant supernatant was subjected to immobilized metal affinity chromatography on nickel-nitrilotriacetic acid (Ni-NTA) resin (Elpis) pre-equilibrated with the lysis buffer. The column was then washed with 10 bed volumes of wash buffer containing 20 mM Tris–HCl (pH 8.0), 500 mM NaCl and 30 mM imidazole. The His$_6$-tagged protein bound to the column was eluted with buffer containing 20 mM Tris–HCl (pH 8.0), 500 mM NaCl, 300 mM imidazole and 5% glycerol. The samples were then further purified through size exclusion chromatography (SEC) on a HiLoad 16/60 Superdex 200 for NTD$_{WT}$ and a Superdex 75 for NTD$_{S92A}$ column (GE Healthcare Life Science) pre-equilibrated with a buffer containing 20 mM Tris–HCl (pH 8.0), 50 mM NaCl, 5% glycerol and 1 mM DTT. The collected fractions containing T4 lysozyme-MCU NTD$_{WT}$ and NTD$_{S92A}$ were concentrated using an Amicon Ultra-15 10 K (Millipore) up to 5.0 mg ml$^{-1}$. The final T4 lysozyme-MCU NTD$_{WT}$ and NTD$_{S92A}$ were stored at −80°C.

MCU NTD-E was purified using a similar procedure with lysis buffer [50 mM Tris–HCl (pH 8.0), 500 mM NaCl, 10 mM imidazole, 5% glycerol, 1 mM PMSF, 1 mM β-mercaptoethanol], wash buffer [50 mM Tris–HCl (pH 8.0), 500 mM NaCl, 40 mM imidazole, 5% glycerol, 1 mM β-mercaptoethanol] and elution buffer [50 mM Tris–HCl (pH 8.0), 500 mM NaCl, 500 mM imidazole, 5% glycerol, 1 mM β-mercaptoethanol]. The samples were then purified using SEC on a HiLoad 16/60 Superdex 75 column (GE Healthcare Life Science) pre-equilibrated with gel filtration buffer [20 mM Tris–HCl (pH 7.8), 100 mM NaCl, 5% glycerol, 1 mM DTT], after which the fractions containing human MCU NTD-E were collected. The protein was then concentrated using an Amicon Ultra-15 10 K (Millipore) to 1.2 mg ml$^{-1}$. The final human MCU NTD-E proteins were stored at −80°C.

### Crystallization of T4 lysozyme-MCU NTD$_{WT}$, T4 lysozyme-MCU NTD$_{S92A}$ and MCU NTD-E

The purified proteins were initially screened for crystallization using the sitting-drop vapour-diffusion method in a 96-well INTELLI-PLATE (Art Robbins Ins.). T4 lysozyme-MCU NTD$_{WT}$ formed needle-shaped crystals after 6 h in reservoir solution containing Index I & II Screen (Hampton Research), 25% (w/v) PEG 3350, 0.2 M ammonium sulphate, 0.1 M Bis-Tris–HCl (pH 6.5) and MCU NTD-E formed needle-shaped crystals after 7 days in a reservoir solution containing SaltRX (Hampton Research), 1.5 M lithium sulphate and 0.1 M Bis-Tris propane (pH 7.0). Additional crystallization trials were performed using the hanging drop vapour-diffusion method. Optimized T4 lysozyme-MCU NTD$_{WT}$ crystals were grown at 20°C in 2-µl drops containing equal volumes of protein and reservoir solution containing 20% PEG 3350, 5% glycerol, 0.3 M ammonium sulphate and 0.1 M Bis-Tris–HCl (pH 5.5). The crystals of T4 lysozyme-MCU NTD$_{S92A}$ were obtained using microseeding method using crystals of T4 lysozyme-MCU NTD$_{WT}$. Once the microcrystals (< 0.01–0.02 mm) of the T4 lysozyme-MCU NTD$_{WT}$ had grown at 20°C, 2-µl of T4 lysozyme-MCU NTD$_{S92A}$ proteins and the 2 µl of reservoir solution were added directly to the 1-µl drops containing T4 lysozyme-MCU NTD$_{WT}$ seed crystals. The final T4 lysozyme-MCU NTD$_{S92A}$ crystals were grown at 20°C in total 5 µl of mixtures containing the WT and the S92A mutant with 1:4 molar ratio. MCU NTD-E crystals were optimized in mother liquor composed of 1.55 M lithium sulphate and 0.1 M Bis-Tris propane (pH 8.0). For data collection, the T4 lysozyme-MCU NTD crystals were directly flash-frozen in liquid nitrogen, while MCU NTD-E crystals were cryoprotected by transferring into a cryoprotectant containing 1.55 M lithium sulphate, 0.1 M Bis-Tris propane (pH 8.0) and 20% glycerol and flash-cooled in liquid nitrogen.

### X-ray diffraction data collection

Diffraction data from T4 lysozyme-MCU NTD$_{WT}$ and MCU NTD-E crystals were collected at 100 K using synchrotron X-ray sources on beamlines 5C and 7A at the Pohang Accelerator Laboratory (PAL, South Korea), NW12A at the Photon Factory (PF) (Tsukuba, Japan) and BL26B1 at Spring-8 (Harima, Japan). Ultimately, we were able to collect the best resolution diffraction data for T4 lysozyme-MCU NTD$_{WT}$ at 1.80 Å resolution and T4 lysozyme-MCU NTD$_{S92A}$ at 2.75 Å, and for MCU NTD-E at 1.50 Å resolution on beamlines 5C and 7A at PAL, South Korea, using single wavelengths (0.9795 and 0.9793 Å, respectively). The crystals belong to the hexagonal space group $P6_5$ ($a = b = 98.1$ Å and $c = 62.4$ Å for T4 lysozyme-MCU NTD$_{WT}$, $a = b = 97.8$ Å and $c = 61.5$ Å for T4 lysozyme-MCU NTD$_{S92A}$, $a = b = 55.5$ Å and $c = 68.9$ Å for MCU NTD-E, with $\alpha = \beta = 90$ degrees and $\gamma = 120$ degrees). Diffraction data were then indexed, processed and scaled using HKL2000 suite [51].

### Structure determination

Initial phases for T4 lysozyme-MCU NTD$_{WT}$ were obtained through MR using Phaser [52] in the CCP4 suite [53] with the structure of bacteriophage T4 lysozyme (PDB code, 2LZM) as the template. As a result, we obtained clear $\sigma_A$-weighted $2F_o–F_c$ maps for the entire MCU NTD structure. All residues in MCU NTD were fitted to the

$\sigma_A$-weighted $2F_o$–$F_c$ maps through model building using Coot [54]. The model was refined using Refmac5 [55], Phenix.refine [56] and Coot [54]. In the final model, $R_{work}$ = 12.7% and $R_{free}$ = 19.0%, and there were no Ramachandran outliers (98.8% most favoured and 1.2% allowed). The phases of the T4 lysozyme-MCU NTD$_{S92A}$ were obtained through MR using Phaser [52] in the CCP4 suite [53] with the structures of the T4 lysozyme (PDB code, 2LZM) and now-solved MCU NTD structure as templates. The model was refined using Refmac5 [55], Phenix.refine [56] and Coot [54]. In the final model, $R_{work}$ = 16.4% and $R_{free}$ = 23.5%, and there were no Ramachandran outliers (97.2% most favoured and 2.8% allowed). R165 of T4 lysozyme-MCU NTD$_{WT}$ and T4 lysozyme-MCU NTD$_{S92A}$ was not modelled in the final structures, because of the weaker electron density map in this residue. The T4 lysozyme-MCU NTD$_{WT}$ and T4 lysozyme-MCU NTD$_{S92A}$ structures included three and six sulphates from mother liquor, respectively.

The phases of the MCU NTD-E structure were obtained through MR using MOLREP [57] in the CCP4 suite [53] with the now-solved MCU NTD structure as a template. All the residues in MCU NTD-E that were missing residues in MCU NTD model were clearly present in the residual electron density maps and were built using Coot [54], and the model was refined using the Refmac5 [55], Phenix.refine [56] and Coot [54]. In the final model, $R_{work}$ = 14.0% and $R_{free}$ = 17.6%, and there were no Ramachandran outliers (99.1% most favoured and 0.9% allowed). Residues 183–185 of MCU NTD-E were not modelled in the final structure because of the weaker electron density map in this region. An unidentified electron density was observed and modelled with a tetraethylene glycol molecule. Extra electron density in the imidazole ring of His170 of MCU NTD-E is modified possibly by unidentified adducts. The statistics of data collection and structure refinement are summarized in Table 1.

## Structural analysis

Superposition of structures was performed using the CCP4 program LSQKAB [58], PyMOL align [59] and Coot SSM superpose [54]. LSQKAB was also used to estimate RMSD (Å) scores for the $C_\alpha$ atoms [58]. The SCOP2 [27], Dali program [28] and CATH database [29] searches were performed using MCU NTD as a template. The Ramachandran statistics were calculated using the program Mol-Probity [60]. The sequence conservation of MCU homologues was calculated using the Consurf server [61]. All molecular graphics were generated using PyMOL version 1.5.0.4 [59].

## Plasmids and constructions

GGECO (green) and RGECO (red) were generous gifts from Robert E. Campbell (Department of Chemistry, University of Alberta, Edmonton, Canada) [36]. For direct targeting into the mitochondrial matrix, a synthetic oligomer corresponding to the N-terminal 31 amino acids of the precursor of subunit VIII of cytochrome C oxidase was fused to the N-terminus of GECO (mito-GGECO).

## Cell cultures and transfection

HeLa and HEK-293 FT cells were maintained in Dulbecco's modified Eagle's medium (GIBCO) supplemented with 10% foetal bovine serum. For the transient expression of the indicated proteins, HeLa and HEK-293 FT cells at 70% confluence were transfected using Fugene HD (Promega) according to the manufacturer's protocol.

## Production of a recombinant lentivirus and generation of stable MCU knockdown (KD) HeLa cells

A recombinant lentivirus expressing shRNA for KD of MCU was produced using previously described methods [42]. The sequence of the shRNA targeting the MCU mRNA was 5′-CAATCAACTCAA GGATGCAAT-3′. After preparation of the recombinant lentivirus, HeLa cells at 70% confluence in a 25-cm$^2$ flask were transduced for 8 h in the presence of 8 g ml$^{-1}$ polybrene, after which the medium was refreshed. 1 μg ml$^{-1}$ puromycin was added to the medium for the selection and maintenance of stable MCU-KD HeLa cells.

## Measurement of mitochondrial membrane potential

HeLa cells cultured on 25-mm coverslips were loaded with 100 nM TMRM (Abcam) for 30 min at 37°C. The coverslips were then placed in a perfusion chamber, and TMRM fluorescence excited at 555 nm was acquired every 3 s using a LSM 700 confocal laser-scanning microscope (arbitrary unit, a.u.) (Carl Zeiss). After 10 baseline acquisitions for 30 s each, 10 μM FCCP (carbonyl cyanide p-trifluoromethoxyphenyl-hydrazone), an uncoupler of oxidation phosphorylation, was rapidly applied to disrupt the mitochondrial membrane potential established by the respiratory chain ($\Delta\Psi$).

## Co-immunoprecipitation assay

Transfected HEK-293 FT cells and HeLa cells were rinsed with cold PBS and solubilized with modified RIPA buffer [20 mM HEPES–NaOH (pH 7.4), 150 mM NaCl, 1 mM EGTA, 1% Triton X-100, 1% NP-40, 1% sodium deoxycholate, 2 mM $Na_3VO_4$, 100 mM NaF, PMSF and protease inhibitor cocktail] for 3 min at 4°C. After centrifugation of the solubilized cells at 13,000 × $g$ for 30 min at 4°C, the supernatant was transferred to new tubes. An anti-GFP antibody (sc-8334, Santa Cruz) or anti-Flag antibody (F-3165, Sigma) was added, and the solution was incubated overnight at 4°C. Protein A-Sepharose CL-4B beads were then used to precipitate the antibodies followed by three washes with solubilizing buffer. After elution of precipitates from the beads with 2× Laemmli sample buffer [65.8 mM Tris–HCl (pH 6.8), 26.3% glycerol, 2.1% SDS, 0.01% bromophenol blue], the elutes were subjected to SDS–PAGE and immunoblotting.

## Confocal imaging

HeLa cells cultured on 25-mm coverslips were co-transfected with plasmids encoding MCU$_{WT}$-GFP/MCU$_{\Delta NTD}$-GFP/mito-GGECO and Mito-DsRed (Clontech). 48 h post-transfection, the cells were imaged using a LSM 700 confocal laser-scanning microscope (Carl Zeiss).

HeLa cells cultured on 25-mm coverslips were transfected with RGECO and mito-GGECO for simultaneous measurement of cytosolic and mitochondrial [$Ca^{2+}$]. 48 h post-transfection, the cells on coverslips were washed with tyrode solution (TS) [140 mM NaCl, 6 mM KCl, 2 mM $CaCl_2$, 1 mM $MgCl_2$, 10 mM glucose and 10 mM

HEPES–NaOH (pH 7.4)] and were placed in a perfusion chamber. After the cells were perfused with fresh TS for 1 min to record baseline data, 100 μM histamine in TS was added. GGECO and RGECO fluorescence excited at 488 and 555 nm, respectively, were simultaneously recorded every 1 s, using a LSM 700 confocal laser-scanning microscope (Carl Zeiss) equipped with a 63× oil immersion objective. The recorded images were analysed and quantified using ZEN 2009 data analysis software (Carl Zeiss). $F_0$ is initial fluorescence intensity. F and $F_{max}$ indicate fluorescence intensity at each time point and maximal fluorescence intensity after the stimulation, respectively. $(F_{max}–F_0)/F_0$ indicates the maximal $Ca^{2+}$ concentration evoked by the stimulation.

### Western blot analysis

Proteins were run on SDS–PAGE and were electrophoretically transferred onto a PVDF membrane. The transferred proteins on the PVDF were incubated with blocking solution containing 5% (w/v) non-fat dried skimmed milk powder and TBST [0.1% Tween-20 in Tris-buffered saline; 137 mM NaCl and 20 mM Tris–HCl, (pH 7.4)] for 1 h at room temperature. Thereafter, the membranes were treated with anti-MCU (HPA016480, Sigma), anti-MCU (raised against a synthetic peptide ($_{328}$NEMDLKRLRDPLQVHLPLRQIG EKDC$_{351}$) from human MCU and named anti-MCU-C), anti-MICU1 (HPA037480, Sigma), anti-Flag (F-3165 and F7425, Sigma), anti-MICU2 (ab101465, Abcam), anti-His (MA1-21315, Thermo Scientific), anti-GFP (sc-9996, Santa Cruz), anti-flavoprotein subunit of complex II (459200, Mito Sciences), anti-myc (sc-40, Santa Cruz) and anti-GAPDH (LF-MA0026, Lab Frontier) antibodies, washed three times with TBST and then incubated with horseradish peroxidase-conjugated secondary antibody (111-035-006 and 115-035-006, Jackson). After washing, the membranes were treated with enhanced chemiluminescence solution (Pierce), and the signals were detected using an ImageQuant LAS 4000 (GE Healthcare Life Sciences).

### Isolation of crude mitochondria and Blue native PAGE (BN–PAGE)

Crude mitochondria were isolated from transfected HEK-293 FT cells and HeLa cells as described previously [8]. Briefly, the cells were rinsed in ice-cold PBS, harvested in isolation buffer [50 mM MOPS, 100 mM KCl, 1 mM EGTA, 5 mM $MgSO_4$, 200 mM sucrose, pH 7.4] and then homogenized by passage of 12 times through a 27.5-guage needle attached to 1-ml syringe. The homogenates were first centrifuged for 10 min at $800 \times g$ to remove nuclei and debris, after which the supernatant was further centrifuged for 10 min at $8,000 \times g$, and the pellet was resuspended with the isolation buffer.

BN–PAGE was performed according to the manufacturer's protocol (Invitrogen). The crude mitochondria were solubilized in BN–PAGE sample buffer containing 1% digitonin for 15 min and then centrifuged at $16,000 \times g$ for 30 min at 4°C. The supernatant was then mixed with BN–PAGE 5% G-250 sample additive to a final concentration of 0.1%. The samples were run on Invitrogen NativePAGE™ Novex® Bis-Tris Gel. After the electrophoresis was completed, the resolved proteins were transferred on to PVDF membranes and probed with anti-Flag (F-3165, Sigma) and anti-MCU antibodies (HPA016480, Sigma).

### Statistics

The experimental values are expressed as means ± SEM. Significance ($P < 0.05$) was determined using the unpaired two-sided Student's $t$-test.

**Expanded View** for this article is available online: http://embor.embopress.org

### Acknowledgements
We thank H. J. Youn (SNU; Seoul, South Korea) and Hye-Yeon Kim (KBSI; Ochang, South Korea) for their support with the data collection and A. G. Murzin (MRC-LMB; Cambridge, United Kingdom) for assistance with identification of new MCU domain. We also thank the staff at the macromolecular X-ray facility of the Korea Basic Science Institute (KBSI; Ochang, South Korea), beamline BL-5C and 7A of the Pohang Accelerator Laboratory (PAL; Pohang, South Korea), beamline NW12A at the Photon Factory (Tsukuba, Japan) and beamline BL26B1 at Spring-8 (Harima, Japan) for their kind help with data collection. This work was supported by grants from the National Research Foundation (NRF) Grants (2013M3A9A7046297, 2013R1A2A2A01068440, 2007-0056157, and 2013R1A1A2062629), "BK21 Plus Program" and "GIST Systems Biology Infrastructure Establishment Grant (2015)", South Korea.

### Author contributions
DHK and SHE planned and organized the experiments. YoL performed purification, crystallization, structure determination, biochemical assays and data analysis. CKM, HKS and YuL performed $Ca^{2+}$ measurements, data analysis and biochemical assays. TGK conducted *in vitro* binding assay. MK and Y-SK generated stable cell lines. JYK carried out gene cloning and expression. YoL, TGK, H-SY, J-GL, JYA, KRP, JJL, JHyK and JW conducted the diffraction experiments and structure determination. CKM, DK and KS performed mammalian cell cultures and gene expression. ZYP performed LC-MS/MS experiments. YoL and JHuK performed bacterial cell cultures and purification. YoL, CKM, JW, DHK and SHE wrote the manuscript.

### Conflict of interest
The authors declare that they have no conflict of interest.

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
