## [Review Process File · EMBO Reports]

Manuscript EMBOR-2015-40436

Structure and function of the N-terminal domain of the human mitochondrial calcium uniporter

Youngjin Lee, Choon Kee Min, Tae Gyun Kim, Hong Ki Song, Yunki Lim, Dongwook Kim, Kahee Shin, Moonkyung Kang, Jung Youn Kang, Hyung-Seop Youn, Jung-Gyu Lee, Jun Yop An, Kyoung Ryoung Park, Jia Jia Lim, Ji Hun Kim, Ji Hye Kim, Zee Yong Park, Yeon-Soo Kim, Jimin Wang, Do Han Kim and Soo Hyun Eom

Corresponding authors: Soo Hyun Eom and Do Han Kim, Gwangju Institute of Science and Technology

Review timeline:	Submission date:	23 March 2015
	Editorial Decision:	14 April 2015
	Revision received:	08 July 2015
	Editorial Decision:	05 August 2015
	Revision received:	06 August 2015
	Accepted:	07 August 2015

Transaction Report:

Editor: Barbara Pauly

1st Editorial Decision 14 April 2015

Thank you very much for the submission of your research manuscript to our editorial office and for your patience while we were waiting to hear back from the referees. We have now received the full set of reviews on your manuscript.

As the detailed reports are pasted below I would prefer not to repeat them here. In essence and while all reviewers agree on the potential interest of the reported structural analysis of the N-terminal domain of MCU, they also feel that the functional and biochemical analyses need to be significantly strengthened before publication of the study. They also all provide suggestions on how to achieve this. In addition, they pinpoint several technical and experimental aspects that need further clarifications.

Overall, and given the reviewers' constructive comments, I would like to give you the opportunity to revise your manuscript, with the understanding that the main concerns of the referees should be addressed. Acceptance of the manuscript will depend on a positive outcome of a second round of review and I should also remind you that it is EMBO reports policy to allow a single round of

revision only and that therefore, acceptance or rejection of the manuscript will depend on the completeness of your responses included in the next, final version of the manuscript.

I look forward to seeing a revised form of your manuscript when it is ready. Should you in the meantime have any questions, please do not hesitate to contact me.

REFEREE REPORTS:

Referee #1:

This is a significant study describing the structure of the MCU N terminus from which the authors suggest several function for this MCU domain. However, the only function that the authors actually show is a very modest reduction in MCU activity upon deletion of the N terminal domain. To increase the significance of the study and bring it to a level of EMBO report publication, the authors should test the validity of some of their predictions. In particular:

- a) Does activation of MCU by CaMKII affected by mutation of S92
- b) Is K180 biotinylated and the biotinylation required for activity

Minor comment: Figure 4 legend and Figure do not match. The results with MCUR1 is shown in the EV, not in this figure. Panel b (not c) shows the multiplex and the traces in c need to be added.

Referee #2:

Lee et al have investigated the structure and function of the N-terminal domain of MCU. MCU is the recently identified pore-forming unit of the mitochondrial Ca²⁺ uniporter. At this point, any structural information on MCU is of relevance.

The structural biology aspect of the study seems to be solid but the functional and some of the biochemical studies are not particularly straightforward. The phenotypes associated with delta NTD MCU overexpression and rescue are relatively mild and the description does not lack overstatements.

Specific comments:

-For the overexpression phenotype (Fig3), the biochemistry was done in HEK cells whereas the functional studies were performed in HeLa. Therefore the relatively mild functional differences, the claimed dominant negative effect of the delta NTD MCU, is not supported effectively by the biochemistry. There is also insufficient technical description in the methods and figure legends, which raises further questions about the Authors interpretations.

-Similar problems also apply to the rescue experiments (Fig4). The difference between the wild type and delta NTD MCU rescue conditions is moderate and assessment of the driving force (deltapsi) and the cytoplasmic calcium signal for each condition is missing.

- Methods lack information about the presentation of the fluorescence values obtained in the calcium and membrane potential measurements. Please explain R, R0, relative intensity etc. Please define what values were used to create the bar charts. From the given information it is not possible to conclude whether the driving force of the calcium uptake was affected.

-Means of statistical analysis used in Figs 3&4 are not described.

-Based on the structural data the functional significance of S92 and K180 is suggested. Functional validation using rescue experiments with point mutants would certainly strengthen the proposal.

Minor:

-The deltaNTD MCU construct seems to be recognized by the anti-MCUab in Fig4A but not in Fig3D.

-P3 "MICU1 and MICU2 are single-pass membrane proteins" Neither MICU1 or MICU2 are considered as transmembrane proteins.

-P8 para2 content of Fig EV6a is not showing what is described in the text

Referee #3:

The paper by Lee and colleagues describe the structure of the N-terminal domain of the human mitochondrial calcium uniporter (MCU), identified in 2011. The paper is of interest given that the crystal structure of full length MCU is still unavailable, even though the findings do not significantly improve the comprehension of the physiological function of MCU. A number of issues should be addressed before acceptance of this work for EMBO reports.

Major points:

- 1) The authors report that MCU NTDs form helical oligomers, suggesting that a crystal packing interface between MCU NTDs is robust. Can this suggestion be extended to a more physiological situation, i.e. to MCU NTDs in solution? The authors should perform biochemical experiments to confirm their hypothesis in solution, in particular for NTD-E.
- 2) The finding that MCU Δ NTD has a dominant-negative effect on calcium uptake is interesting but also surprising. Since a dominant-negative effect has been reported for MCU^b which has the pore region altered with respect to MCU, a more direct proof should be provided on how the lack of NTD exerts its effect on MCU activity. For example, does MCU Δ NTD form calcium-permeable channels with lower conductance or lower open probability than wt? This important point can be easily tested by electrophysiology. Alternatively, might it be possible that only a part of the MCU Δ NTD folds correctly and give rise to functional channels? It is difficult to explain the dominant-negative effect on the basis of the provided data.
- 3) The resolution of the dataset reported in Table I. does not correspond to an acceptable limit. Data should be cut according to a more reasonable Rmerge value in the highest resolution shell (e.g. max 50-60%). I/ σ / is also indicative of an actual lower resolution.
- 4) Discussion is very speculative especially the part regarding the eventual role of the NTD domain in the clustering of unicomplexes at microdomains. If the authors decide to maintain this part, they should clearly state that these are pure assumptions. The hypotheses regarding modification of MCU activity by specific phosphorylation etc. sites in the NTD domain could be easily tested using site-directed mutagenesis and should be done at least for a few sites.
- 5) It seems that when performing IP experiments, the authors did not introduce washing steps before elution with the solubilizing buffer. If it was so, the IPs would not be representative of real interactions.

Minor points:

- 1) When citing MCUR1, the authors should mention already in the introduction section that this protein has been proposed to have a function different from that of modulating MCU activity (Paupe et al, 2015).
- 2) The tetrameric nature of MCU channel is indicated also by biochemical experiments, not only by molecular modeling (Raffaello et al, 2013).
- 3) In addition to mentioning the role of MCU for apoptosis, the authors should refer to reviews illustrating the importance of MCU for metabolism as well and should briefly discuss the phenotype of MCU knock-out organisms.
- 4) Please insert page numbers.
- 5) Language errors are present and the sentence "In addition, our results suggest a possibility of lipid-binding...." in the Discussion section should be re-written, since it is not clear.

1st Revision - authors' response

08 July 2015

Response to referees' comments

We would like to thank the editor and the referees for their constructive comments on our manuscript entitled "**Structure and function of the N-terminal domain of the human mitochondrial calcium uniporter**" submitted to EMBO reports. For revision of the manuscript, we have conducted numerous new experiments (e.g. functional and structural characterization of the potential phosphorylation site on S92 and biotinylation site on K180 of MCU by site-directed mutagenesis). The new data are now shown in the revised manuscript in detail. We are grateful for

your valuable suggestions, and we hope you and the referees now find the manuscript suitable for publication.

Referee #1 (Remarks to the Author):

This is a significant study describing the structure of the MCU N terminus from which the authors suggest several function for this MCU domain. However, the only function that the authors actually show is a very modest reduction in MCU activity upon deletion of the N terminal domain. To increase the significance of the study and bring it to a level of EMBO report publication, the authors should test the validity of some of their predictions. In particular:

a) Is activation of MCU by CaMKII affected by mutation of S92?

Response: Thank you for this pertinent suggestion. We examined the effects of the S92 mutation (MCU_{S92A}) on the structure and function of MCU (page 11–12; Fig 5 and 6). When we monitored mitochondrial Ca²⁺ uptake in MCU_{S92A}-rescued MCU-KD HeLa cells, mitochondrial Ca²⁺ uptake was impaired (Fig 5C and 5D). We also determined the crystal structure of the MCU NTD_{S92A} structure at 2.75 Å resolution. We observed conformational changes in the conserved L2-L4 loops from MCU NTD_{S92A} (page 12, lines 4–19; Fig 6; Table I), suggesting that these structural changes lead to impairment of the mitochondrial Ca²⁺ uptake. Based on these results, we predict that S92 phosphorylation induces conformational changes and charge distribution in L2-L4 loops, which is expected to be important for the modulation of the MCU function.

b) Is K180 biotinylated and the biotinylation required for activity?

Response: Recent studies have reported that K180 in MCU is biotinylated [Schiapparelli *et al.* (2014) *J Proteome Res*, **13**: 3966–3978 (reference 26)]. To determine whether biotinylation is required for activity, we examined mitochondrial Ca²⁺ uptake in MCU_{K180A}-rescued MCU-KD HeLa cells (page 11, line 6–page 12, line 2; Fig 5). Our results indicated that mitochondrial Ca²⁺ uptake was not altered by the mutation, suggesting that biotinylation of K180 does not affect mitochondrial Ca²⁺ uptake activity.

Minor comment: Figure 4 legend and Figure do not match. The results with MCUR1 is shown in the EV, not in this figure. Panel b (not c) shows the multiplex and the traces in c need to be added.

Response: We apologize for these oversights. Per your suggestion, we replaced the images for Fig 4 and revised legend. The legend and figure now correspond.

Referee #2 (Remarks to the Author):

Lee et al have investigated the structure and function of the N-terminal domain of MCU. MCU is the recently identified pore-forming unit of the mitochondrial Ca²⁺ uniporter. At this point, any structural information on MCU is of relevance. The structural biology aspect of the study seems to be solid but the functional and some of the biochemical studies are not particularly straightforward. The phenotypes associated with delta NTD MCU overexpression and rescue are relatively mild and

the description does not lack overstatements.

Specific comments:

-For the overexpression phenotype (Fig3), the biochemistry was done in HEK cells whereas the functional studies were performed in HeLa. Therefore the relatively mild functional differences, the claimed dominant negative effect of the delta NTD MCU, is not supported effectively by the biochemistry. There is also insufficient technical description in the methods and figure legends, which raises further questions about the Authors interpretations.

Response: Per your suggestion, we performed the biochemical study using HeLa cells. These new data from HeLa cells were consistent with the previous study of HEK293 FT cells. These new data are now presented in the revised manuscript (page 8, line 22–page 9, line 12; Fig 3, Fig 4, and Fig EV4E) and the previous results from HEK293 FT cells are also included in Appendix Fig 5. We added the details of biochemical and functional methods to the **Materials and Methods** and the legend for Fig 3 .

-Similar problems also apply to the rescue experiments (Fig4). The difference between the wild type and delta NTD MCU rescue conditions is moderate and assessment of the driving force (deltapsi) and the cytoplasmic calcium signal for each condition is missing.

Response: The cytoplasmic Ca^{2+} signal and mitochondrial membrane potential data for each condition are now shown in Fig 4E–4H, and described in the revised text (page 10, lines 17–18; page 11, lines 2–3).

- Methods lack information about the presentation of the fluorescence values obtained in the calcium and membrane potential measurements. Please explain R, R0, relative intensity etc. Please define what values were used to create the bar charts. From the given information it is not possible to conclude whether the driving force of the calcium uptake was affected.

Response: The missing information has now been added to the **Materials and Methods** (page 21, line 23–page 22, line 3) and also the legends for Fig 3–5. The detailed experimental procedure for measurements of mitochondrial membrane potential has now been added to the **Materials and Methods** (page 22, lines 5–12) and legends for Fig 3 and 4.

-Means of statistical analysis used in Figs 3&4 are not described.

Response: Thank you for pointing this out. Means of the statistical analysis are now shown in Fig 3 and 4 and described in the figure legends.

-Based on the structural data the functional significance of S92 and K180 is suggested. Functional validation using rescue experiments with point mutants would certainly strengthen the proposal.

Response: We thank the referee for this pertinent suggestion. We investigated the mitochondrial Ca^{2+} uptake in MCU_{S92A} or $\text{MCU}_{\text{K180A}}$ -rescued MCU-KD HeLa cells. Our results showed that mitochondrial Ca^{2+} uptake was impaired in MCU_{S92A} -rescued cells, but not in $\text{MCU}_{\text{K180A}}$ -rescued

cells (page 11, line 6–page 12, line 2; Fig 5). We also determined the crystal structure of the MCU NTD_{S92A} structure at 2.75 Å resolution. We observed conformational changes in the conserved L2-L4 loops from MCU NTD_{S92A} (page 12, lines 4–19; Fig 6; Table I), suggesting that these structural changes lead to impairment of the mitochondrial Ca²⁺ uptake.

Minor:

-The deltaNTD MCU construct seems to be recognized by the anti-MCUab in Fig4A but not in Fig3D.

Response: We thank the referee for pointing out the discrepancy in the detection ability of the antibodies. In fact, the antibody used for the study shown in Fig 3D was obtained from Sigma Co. Due to its epitope (residues 47–152 of MCU, mostly in the NTD region), it cannot detect MCU_{ΔNTD}. The antibody used for the study (shown in Fig 4A) was generated in our lab. Since the epitope is the C-terminal region (₃₂₈NEMDLKRLRDPLQV HLPLRQIGEKDC₃₅₁), it can detect both MCU_{WT} and MCU_{ΔNTD}. For greater clarity, detailed information on both antibodies has now been included in the **Materials and Methods** (page 23, lines 15–17) and the legend for Fig 4.

-P3 "MICU1 and MICU2 are single-pass membrane proteins" Neither MICU1 or MICU2 are considered as transmembrane proteins.

Response: Per your suggestion, the description of MICU1 and MICU2 was re-written in the text (page 3, lines 8–9).

-P8 para2 content of Fig EV6a is not showing what is described in the text

Response: Thank you for pointing this out. We revised the text (page 8, lines 12–14) as follows: “Subcellular localization of MCU_{ΔNTD} within the mitochondria was confirmed by the co-expression of C-terminally GFP-tagged MCU_{ΔNTD} and DsRed-Mito (Fig EV4C).”

Referee #3 (Remarks to the Author):

The paper by Lee and colleagues describe the structure of the N-terminal domain of the human mitochondrial calcium uniporter (MCU), identified in 2011. The paper is of interest given that the crystal structure of full length MCU is still unavailable, even though the findings do not significantly improve the comprehension of the physiological function of MCU. A number of issues should be addressed before acceptance of this work for EMBO reports.

Major points:

1) The authors report that MCU NTDs form helical oligomers, suggesting that a crystal packing interface between MCU NTDs is robust. Can this suggestion be extended to a more physiological situation, i.e. to MCU NTDs in solution? The authors should perform biochemical experiments to confirm their hypothesis in solution, in particular for NTD-E.

Response: To examine whether MCU NTD-E oligomerizes in solution, we performed an *in vitro* cross-linking assay using glutaraldehyde in phosphate buffered saline (PBS) or in the crystallization solution. We found that MCU NTD-Es were cross-linked by glutaraldehyde in phosphate buffered saline (PBS) or in the crystallization solution, suggesting that MCU NTD-E is required for MCU oligomerization (Page 8, lines 1–3; Appendix Fig 3). We included details of these methods in **Appendix Materials and Methods**.

2) The finding that $MCU_{\Delta NTD}$ has a dominant-negative effect on calcium uptake is interesting but also surprising. Since a dominant-negative effect has been reported for MCU_b which has the pore region altered with respect to MCU, a more direct proof should be provided on how the lack of NTD exerts its effect on MCU activity. For example, does $MCU_{\Delta NTD}$ form calcium-permeable channels with lower conductance or lower open probability than wt? This important point can be easily tested by electrophysiology. Alternatively, might it be possible that only a part of the $MCU_{\Delta NTD}$ folds correctly and give rise to functional channels? It is difficult to explain the dominant-negative effect on the basis of the provided data.

Response: We thank the referee for his/her concern on the functional role of NTD. Since the suggested electrophysiological study was not possible in the time available to us for revision, we adopted the alternative approach to check whether only a part of the $MCU_{\Delta NTD}$ folds correctly to give rise to functional channels. Our CD analysis of $MCU_{\Delta NTD}$ revealed the canonical α -helical pattern and the α -helical content matched well with that of the secondary structure that was predicted computationally, suggesting that $MCU_{\Delta NTD}$ folded properly. The results are now described in the text (page 8, lines 17–21; Appendix Fig 4A and 4B).

3) The resolution of the dataset reported in Table I. does not correspond to an acceptable limit. Data should be cut according to a more reasonable Rmerge value in the highest resolution shell (e.g. max 50-60%). I/σ is also indicative of an actual lower resolution.

Response: We re-processed the diffraction data and re-refined the structures of the T4 lysozyme-MCU NTD (1.80 Å) and MCU NTD-E (1.50 Å), showing improved data collection and refinement statistics. In addition, we determined the crystal structure of MCU NTD S92A mutant in T4 lysozyme fusion as well (2.75 Å). We have now revised Table I according to the final structural statistics and the additional information has been added to the **Materials and Methods**.

4) Discussion is very speculative especially the part regarding the eventual role of the NTD domain in the clustering of unicomplexes at microdomains. If the authors decide to maintain this part, they should clearly state that these are pure assumptions. The hypotheses regarding modification of MCU activity by specific phosphorylation etc. sites in the NTD domain could be easily tested using site-directed mutagenesis and should be done at least for a few sites.

Response: (a) The text was re-written significantly to eliminate any ambiguity, per your suggestion (page 13 lines 13–19).

(b) We performed experiments on mitochondrial Ca^{2+} uptake in MCU_{S92A} or $\text{MCU}_{\text{K180A}}$ -rescued MCU-KD HeLa cells. The mitochondrial Ca^{2+} uptake was only impaired in MCU_{S92A} -rescued cells, but not in $\text{MCU}_{\text{K180A}}$ -rescued cells (page 11 line 6–page 12 line 2; Fig 5). We also determined the crystal structure of the MCU NTD_{S92A} structure at 2.75 Å resolution. We observed conformational changes in the conserved L2-L4 loops from MCU NTD_{S92A} (page 12 lines 4–19; Fig 6; Table I), suggesting that these structural changes lead to impairment of the mitochondrial Ca^{2+} uptake.

5) *It seems that when performing IP experiments, the authors did not introduce washing steps before elution with the solubilizing buffer. If it was so, the IPs would not be representative of real interactions.*

Response: As described in **Materials and Methods** (Page 22, lines 15–18, 22), we introduced the washing steps (3 times with the solubilization buffer) for the all IP experiments.

Minor points:

1) *When citing MCUR1, the authors should mention already in the introduction section that this protein has been proposed to have a function different from that of modulating MCU activity (Paupe et al, 2015).*

Response: As suggested, this new sentence has now been added to the introduction (page 3, lines 11–13).

2) *The tetrameric nature of MCU channel is indicated also by biochemical experiments, not only by molecular modeling (Raffaello et al, 2013).*

Response: Per your suggestion, the text was revised as shown in page 3, lines 17–19.

3) *In addition to mentioning the role of MCU for apoptosis, the authors should refer to reviews illustrating the importance of MCU for metabolism as well and should briefly discuss the phenotype of MCU knock-out organisms.*

Response: Per your suggestion, two paragraphs including importance of MCU for apoptosis and metabolism and the description of phenotype of MCU knock-out organism are added (page 3, line 22–page 4, line 9).

4) *Please insert page numbers.*

Response: Thank you for this suggestion. Page and line numbers have been added to the text.

5) *Language errors are present and the sentence "In addition, our results suggest a possibility of lipid-binding..." in the Discussion section should be re-written, since it is not clear.*

Response: We eliminated this sentence in the text, because it was not necessary.

Thank you for your patience while we have reviewed your revised manuscript. As you will see from the reports below, the referees are now all positive about its publication in EMBO reports. I am therefore writing with an 'accept in principle' decision, which means that I will be happy to accept your manuscript for publication once a few minor issues/corrections have been addressed, as outlined in the report of referee 2.

If all remaining corrections have been attended to, you will then receive an official decision letter from the journal accepting your manuscript for publication in the next available issue of EMBO reports. This letter will also include details of the further steps you need to take for the prompt inclusion of your manuscript in our next available issue.

Thank you for your contribution to EMBO reports.

REFEREE REPORTS:

Referee #1:

The authors did an excellent job in addressing the concerns pointed out by the reviewers and the m/s is now suitable for publication.

Referee #2:

The revised MS by Lee et al addressed all the major concerns raised in the reviews. However, the current draft still contains numerous issues that require the Authors' attention. Please address the below listed concerns and carefully check the entire MS:

Pg13 ln8: "we identified that MCU-NTD directly interacts with MCUR1" Only co-IP is presented in FigEV5 which is not evidence for a "direct" interaction.

It remains unclear to this reviewer what statistical test was used to evaluate the calcium responses in Figs3-5.

Pg3 ln 23: neither ref 14,15 nor other papers provided evidence that intramitochondrial [Ca²⁺] is relevant for fusion or fission.

Pg22 ln 6: Cells were loaded with "10µm" TMRM.

Pg23 ln4-8: It is unclear why "Confocal Imaging" is separated from the section on calcium measurements on pg21.

Pg38 ln1-2: Meaning of "TMRE loading...." sentence is unclear.

Figure EV5: Source of reaction with anti-myc ab in lane 1 is unclear since no transfection with a myc-tagged construct is indicated.

Referee #3:

The authors addressed all my concerns and added new experiments, rendering the paper more sound. Therefore I recommend its publication in EMBO Reports.

Response to referees' comments

We thank the editor and the referees for their constructive reviews. Our responses to the minor comments of Referee #2 are listed below.

Referee #2 (Remarks to the Author):

The revised MS by Lee et al addressed all the major concerns raised in the reviews. However, the current draft still contains numerous issues that require the Authors' attention. Please address the below listed concerns and carefully check the entire MS:

Pg13 ln8: "We identified that MCU-NTD directly interacts with MCUR1" Only co-IP is presented in FigEV5 which is not evidence for a "direct" interaction.

Response: For evidence of a direct interaction, we included the *in vitro* pull-down results in Fig. EV5B (it was shown in the previous version already).

It remains unclear to this reviewer what statistical test was used to evaluate the calcium responses in Figs3-5.

Response: The Student's t-test was used for statistics. The information is now shown in the revised text (page 24, line 17–19).

Pg3 ln 23: neither ref 14,15 nor other papers provided evidence that intramitochondrial [Ca²⁺] is relevant for fusion or fission.

Response: Per your suggestion, a new reference (#14) to address the issue was replaced previous reference (#14) in "References" and previous reference (#14) was eliminated in "References", because it was not necessary.

Pg22 ln 6: Cells were loaded with "10µm" TMRM.

Response: It was corrected to "100 nM" (page 21, line 14).

Pg23 ln4-8: It is unclear why "Confocal Imaging" is separated from the section on calcium measurements on pg21.

Response: Per your suggestion, the two paragraphs are now combined (page 22, line 18–page 23, line 7)

Pg38 ln1-2: Meaning of "TMRE loading...." sentence is unclear.

Response: The new amended sentence is: "The value of TMRM loading (Relative intensity) was calculated by dividing each fluorescence intensity of TMRM measured with MCU_{WT} or MCU_{ΔNTD} vector by that of TMRM measured with empty vector" It is now shown in page 38, line 1–3 and page 39, line 2–5.

Figure EV5: Source of reaction with anti-myc ab in lane 1 is unclear since no transfection with a myc-tagged construct is indicated.

Response: We corrected it by replacing “-“ with “+”.

3rd Editorial Decision

07 August 2015

I am very pleased to accept your manuscript for publication in the next available issue of EMBO reports.

Thank you for your contribution to EMBO reports and congratulations on a successful publication. Please consider us again in the future for your most exciting work.